# A Multi-modal Global Instance Tracking Benchmark (MGIT): Better Locating Target in Complex Spatio-temporal and Causal Relationship

**Shiyu Hu**[1,2]   **Dailing Zhang**[1,2]   **Meiqi Wu**[3]   **Xiaokun Feng**[1,2]
**Xuchen Li**[4]   **Xin Zhao**[1,2]   **Kaiqi Huang**[1,2,5]

[1]School of Artificial Intelligence, University of Chinese Academy of Sciences
[2]Institute of Automation, Chinese Academy of Sciences
[3]School of Computer Science and Technology, University of Chinese Academy of Sciences
[4]School of Computer Science, Beijing University of Posts and Telecommunications
[5]Center for Excellence in Brain Science and Intelligence Technology, Chinese Academy of Sciences
{hushiyu2019, zhangdailing2023, fengxiaokun2022}@ia.ac.cn, wumeiqi18@mails.ucas.ac.cn,
xuchenli@bupt.edu.cn, {xzhao,kqhuang}@ia.ac.cn

## Abstract

Tracking an arbitrary moving target in a video sequence is the foundation for high-level tasks like video understanding. Although existing visual-based trackers have demonstrated good tracking capabilities in short video sequences, they always perform poorly in complex environments, as represented by the recently proposed global instance tracking task, which consists of longer videos with more complicated narrative content. Recently, several works have introduced natural language into object tracking, desiring to address the limitations of relying only on a single visual modality. However, these selected videos are still short sequences with uncomplicated spatio-temporal and causal relationships, and the provided semantic descriptions are too simple to characterize video content. To address these issues, we (1) first propose a new **m**ulti-modal **g**lobal **i**nstance **t**racking benchmark named **MGIT**. It consists of 150 long video sequences with a total of 2.03 million frames, aiming to fully represent the complex spatio-temporal and causal relationships coupled in longer narrative content. (2) Each video sequence is annotated with three semantic grains (*i.e.*, *action*, *activity*, and *story*) to model the progressive process of human cognition. We expect this **multi-granular annotation strategy** can provide a favorable environment for multi-modal object tracking research and long video understanding. (3) Besides, we execute comparative experiments on existing multi-modal object tracking benchmarks, which not only explore the impact of different annotation methods, but also validate that our annotation method is a feasible solution for coupling human understanding into semantic labels. (4) Additionally, we conduct detailed experimental analyses on MGIT, and hope the explored performance bottlenecks of existing algorithms can support further research in multi-modal object tracking. The proposed benchmark, experimental results, and toolkit will be released gradually on http://videocube.aitestunion.com/.

## 1 Introduction

Single object tracking (SOT) is an important computer vision task that aims to locate an arbitrary moving target in a video sequence, and can be regarded as the foundation for high-level tasks like video understanding. In the past decade, researchers have proposed numerous high-quality benchmarks [4, 5, 6, 7, 8, 9] for the visual-based SOT task, and a series of trackers [10, 11, 12, 13, 14,

37th Conference on Neural Information Processing Systems (NeurIPS 2023) Track on Datasets and Benchmarks.

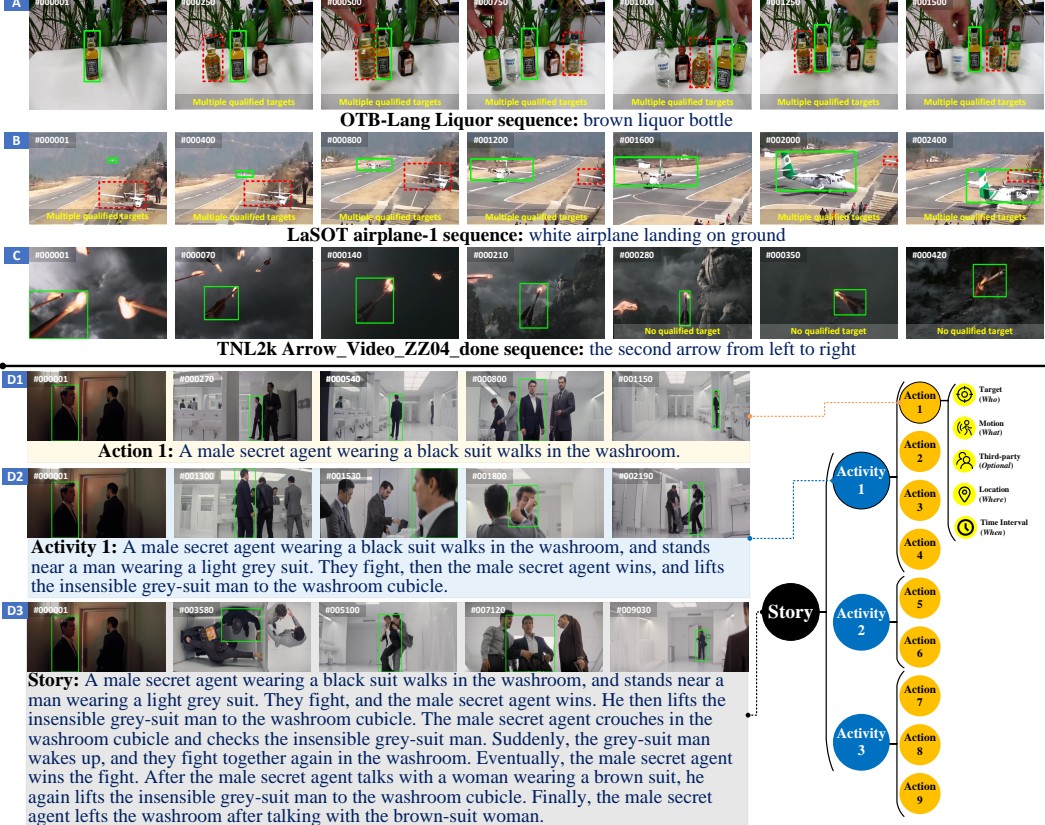

Figure 1: Comparison of MGIT and other multi-modal object tracking benchmarks. (A-C) Examples of video content and semantic descriptions on OTB-Lang [1], LaSOT [2], and TNL2K [3]. The green bounding box (BBox) indicates ground truth, while the red dashed BBox indicates other objects that satisfy the semantic description. These benchmarks have short sequences with simple narrative content. Besides, their semantic labels mainly describe the first frame, which may misguide algorithms. (D1-D3) An example of the multi-granular annotation strategy used by MGIT. Compared to existing benchmarks, MGIT contains longer sequences with more complex narratives, and the multi-granular annotation provides more prosperous and flexible information to portray long videos.

15, 16] have demonstrated good tracking capabilities in these environments, especially in short video sequences ranging from hundreds to thousands of frames. However, researchers noticed that most trackers always perform poorly in longer videos with more complicated narrative content. Besides, only relying on a single visual modality also limits the application scenarios. Thus, several works have begun to offer additional semantic annotations for SOT task.

As the first multi-modal SOT benchmark, OTB-Lang [1] provides a language description for the classic OTB [5] benchmark, hoping to provide a more natural human-machine interaction method. The long-term tracking benchmark LaSOT [2, 17] also supplies a semantic annotation for each sequence, desiring to utilize linguistic features to improve the tracking performance. TNL2k [3] wants to achieve more flexible and accurate tracking ability with more explicit information (*e.g.*, location information) in the semantic description. Although these multi-modal benchmarks have introduced semantic information into visual object tracking, they still face the following problems. (1) **Short sequences with uncomplicated spatio-temporal and causal relationships**: Existing works mainly focus on videos with hundreds to thousands of frames (the average sequence lengths of OTB-Lang, LaSOT, and TNL2k are 590 frames, 2,502 frames, and 622 frames), while shorter video sequences are always insufficient to reflect complex narrative content. (2) **Simple semantic descriptions**: The quality of semantic information is critical to multi-modal trackers' performance, while incorrect or ambiguous semantic information may misguide algorithms in tracking interference [18]. However, the semantic labels in existing works mainly describe the state in the first frame, but lack the portrayal

of the complete sequence. For example, the *brown liquor bottle* description of Figure 1 (A) cannot distinguish the object from the interference (another brown liquor bottle). In Figure 1 (B), *white airplane landing on ground* may also misdirect trackers to locate another airplane that has parked on the right ground. In Figure 1 (C), *the second arrow from left to right* only satisfies to represent the object state at the beginning of the sequence; as the object moves, the position constraint contained in the semantic information will become misleading. Consequently, a better way to construct a multi-modal benchmark is not to provide a simple natural language description for short videos, but to design a scientific way to couple human understanding of long videos into semantic labels.

Therefore, we should first select suitable long videos with rich narrative relationships to compose a complex environment. VideoCube [19] is a high-quality benchmark recently released for the global instance tracking (GIT) task (*i.e.*, search an arbitrary user-specified instance in a video without any assumptions about motion consistency), which can be regarded as expanding the definition of traditional SOT task (*i.e.*, tracking a target in single-camera and single-scene) to success model the human visual tracking ability in a complex environment. Thus, we selected 150 representative long video sequences from VideoCube to form a new multi-modal benchmark named **MGIT**. The proposed new benchmark is consistent with the distribution of the original VideoCube in all dimensions (*e.g.*, length, scene categories, object classes, motion modes, spatio-temporal consistency, and difficulty). Besides, we carefully check the content of each sequence to ensure that the selected data contain as many different types of video narratives as possible. Figure 1 (D1-D3) illustrates an example in MGIT. Compared with other works, sequences in MGIT include more complex content (*i.e.*, the spatial-temporal variation and causal relationship are more complicated).

Besides, we design a **multi-granular annotation strategy** to provide scientific natural language information. On the one hand, existing research has indicated that complex narrative content can be perceived as several components and their relations, which is consistent with cognitive intuition [20]. On the other hand, the process of human comprehension and cognitive development is progressive as well [21, 22]. Therefore, designing a hierarchical structure to represent the video content is a reasonable annotation method. As shown in Figure 1 (D1-D3), each sequence in MGIT is annotated with three semantic grains (*i.e.*, *action*, *activity*, and *story*). We hope this method can provide a step-by-step "learning" environment for multi-modal trackers, in which they can first learn multi-modal information at a fine-grained level (*action*), then gradually develop to a morea comprehensive level (*activity*), and finally understand the complex video narrative at a *story* level like humans.

**Contributions.** (1) We propose a new multi-modal benchmark named MGIT. It consists of 150 long videos with a total of 2.03 million frames, and the average length of a single sequence is *5~ 22 times longer* than existing multi-modal benchmarks. We hope this new benchmark fully represents the complex spatio-temporal and causal relationships coupled in longer narrative content (Section 3.1). (2) We design a multi-granular annotation strategy for providing scientific semantic information. Via this strategy, MGIT can provide a favorable environment for multi-modal object tracking research and long video understanding (Section 3.2). (3) We execute comparative experiments on other benchmarks. Experimental results explore the impact of different annotation methods, and validate that the proposed strategy is a feasible solution for coupling human understanding into semantic labels (Section 4.2). (4) We conduct detailed experimental analyses on MGIT. Results indicate that existing methods still have significant room for improvement in multi-modal tracking (Section 4.3). The proposed benchmark, experimental results, and toolkit will be released gradually on http://videocube.aitestunion.com/.

## 2 Related Work

**Benchmarks with Visual Information.** Standard SOT trackers are always initialized in the first frame by a target's bounding box (BBox), then continuously locating it in the video sequence. Since 2013, many benchmarks represented by OTB [4, 5] and VOT [6, 23] have been released, and these standardized datasets with scientific evaluation mechanisms promote the SOT research. With the development of deep learning techniques, these short-term and small-scale benchmarks have struggled to support data-driven trackers. Thus, several researchers have started to design larger-scale datasets like GOT-10k [9] and TrackingNet [8], while others have tried to collect data with longer videos and proposed long-term tracking benchmarks like OxUvA [24] and VOT-LT [25, 26]. Recently, some researchers have noticed that short-term and long-term tracking tasks include a continuous motion assumption in their definitions, resulting in the experimental environments being restricted to

single-camera and single-scene. Therefore, they propose the global instance tracking task [19] with a new benchmark named VideoCube to track an arbitrary moving target in any type of video.

**Benchmarks with Visual and Semantic Information.** Unlike numerous visual benchmarks that have evolved over a decade, multi-modal benchmarks combining visual and semantic information have only received attention lately. OTB-Lang [1] is the first multi-modal SOT benchmark, which provides additional natural language description for sequences in OTB100 [5] benchmark. However, the limited dataset scale has prevented the multi-modal SOT task from receiving widespread attention. After that, a large-scale and long-term tracking benchmark LaSOT [2, 17] is released with multi-modal annotations. In the same year, researchers propose the TNL2k [3] benchmark to achieve more flexible and accurate object tracking with natural language. These two benchmarks have provided a prosperity of data and have facilitated the generation of various multi-modal trackers.

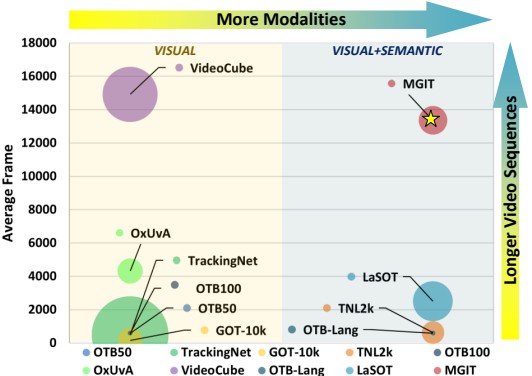

Figure 2: Comparison between MGIT with other SOT benchmarks, including visual-based (*e.g.*, OTB50 [4], OTB100 [5], GOT-10k[9], TrackingNet [8], OxUvA [24], and VideoCube [19]) and multi-modal SOT benchmarks (*e.g.*, OTB-Lang [1], LaSOT [2], and TNL2k [3]). The bubble diameter is in proportion to the total frames of a benchmark, and the vertical coordinate represents the average sequence length of each benchmark. Obviously, the proposed MGIT includes *longer videos* with *multi-modal information*.

As shown in Figure 2, existing works either focus on visual modality, or concentrate on multi-modality but lack longer videos with complex content. Besides, Figure 1 indicates a more scientific annotation strategy is also needed for providing high-quality semantic information. These limitations prompt us to propose MGIT, hoping to construct a more complex and flexible environment for research.

**Algorithms with Bounding Box.** Visual-based trackers always utilize the target's appearance and motion information to accomplish the tracking process, including the correlation filter (CF) based trackers [27, 28], Siamese neural network (SNN) based trackers [29, 30, 31, 32, 33, 34, 11, 35, 36, 10], the combination of CF and SNN [37, 38, 12, 13], and the transformer-based trackers [39, 14, 15, 16]. Before 2021, SNN-based trackers are the prevalent methods. Recently, transformer-based trackers have demonstrated exemplary performance and gradually become the dominant architecture.

**Algorithms with Bounding Box and Natural Language.** Tracking a moving target with visual and semantic information is a new task for SOT research; thus, representative works are mainly released in recent two years. AdaSwitcher [3] is released with the TNL2k benchmark, which proposes a switcher that utilizes natural language to alternate search mechanism (*i.e.*, switch between the global search visual grounding module and the local visual tracking module). GTI [40] decomposes the visual language tracking task into three sub-tasks: tracking, grounding, and integration, and verifies the performance of each sub-module. SINT [41] proposes a semantic information fusion module that can be utilized across various SNN-based trackers. VLT [42] introduces a modality mixer named ModaMixer with asymmetric ConvNet search, which aims to demonstrate pure ConvNet models can achieve comparable results to state-of-the-art (SOTA) transformer-based algorithms. Besides, the proposed ModaMixer can further improve performance when directly applied to transformer-based trackers. JointNLT [18] unifies visual grounding and tracking as a coherent task (*i.e.*, locating referred objects based on visual-language references). It employs the transformer-based architecture to model the relation between natural language and visual information.

## 3 Construction of MGIT

We propose a new multi-modal benchmark named **MGIT** and design a **multi-granular annotation strategy** for generating scientific semantic information. On the one hand, we have carefully selected 150 longer video sequences to form MGIT (please refer to Section A.2 in the Appendix for more details), hoping this complex environment can promote visual tracking and video understanding research. On the other hand, we hope this multi-granular annotation strategy can provide a step-by-

step "learning" environment for multi-modal trackers. Like humans can increase their comprehension by gradually increasing the learning difficulty, trackers can first learn at a fine-grained level (*action*), then to a more comprehensive level (*activity*), and finally accomplish a *story* level understanding of long video sequences. A well-trained elite annotation team is selected to execute this task instead of crowdsourcing, and the annotation quality is ensured through various efforts. The detailed workflow has been outlined in Section A.3.2 of the Appendix.

## 3.1 Data Collection

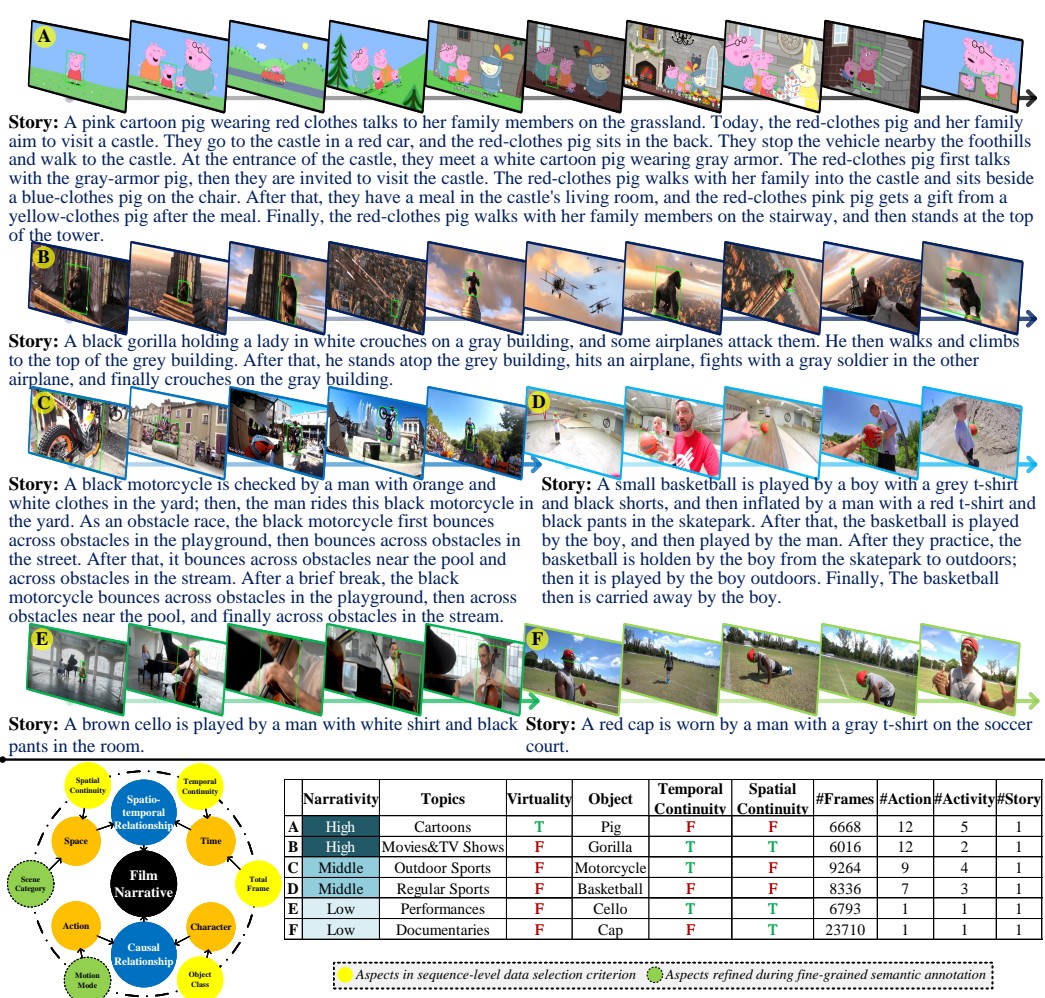

**Story:** A pink cartoon pig wearing red clothes talks to her family members on the grassland. Today, the red-clothes pig and her family aim to visit a castle. They go to the castle in a red car, and the red-clothes pig sits in the back. They stop the vehicle nearby the foothills and walk to the castle. At the entrance of the castle, they meet a white cartoon pig wearing gray armor. The red-clothes pig first talks with the gray-armor pig, then they are invited to visit the castle. The red-clothes pig walks with her family into the castle and sits beside a blue-clothes pig on the chair. After that, they have a meal in the castle's living room, and the red-clothes pink pig gets a gift from a yellow-clothes pig after the meal. Finally, the red-clothes pig walks with her family members on the stairway, and then stands at the top of the tower.

**Story:** A black gorilla holding a lady in white crouches on a gray building, and some airplanes attack them. He then walks and climbs to the top of the grey building. After that, he stands atop the grey building, hits an airplane, fights with a gray soldier in the other airplane, and finally crouches on the gray building.

**Story:** A black motorcycle is checked by a man with orange and white clothes in the yard; then, the man rides this black motorcycle in the yard. As an obstacle race, the black motorcycle first bounces across obstacles in the playground, then bounces across obstacles in the street. After that, it bounces across obstacles near the pool and across obstacles in the stream. After a brief break, the black motorcycle bounces across obstacles in the playground, then across obstacles near the pool, and finally across obstacles in the stream.

**Story:** A small basketball is played by a boy with a grey t-shirt and black shorts, and then inflated by a man with a red t-shirt and black pants in the skatepark. After that, the basketball is played by the boy, and then played by the man. After they practice, the basketball is holden by the boy from the skatepark to outdoors; then it is played by the boy outdoors. Finally, The basketball then is carried away by the boy.

**Story:** A brown cello is played by a man with white shirt and black pants in the room.

**Story:** A red cap is worn by a man with a gray t-shirt on the soccer court.

| | Narrativity | Topics | Virtuality | Object | Temporal Continuity | Spatial Continuity | #Frames | #Action | #Activity | #Story |
|---|---|---|---|---|---|---|---|---|---|---|
| A | High | Cartoons | T | Pig | F | F | 6668 | 12 | 5 | 1 |
| B | High | Movies&TV Shows | F | Gorilla | T | T | 6016 | 12 | 2 | 1 |
| C | Middle | Outdoor Sports | F | Motorcycle | T | F | 9264 | 9 | 4 | 1 |
| D | Middle | Regular Sports | F | Basketball | F | F | 8336 | 7 | 3 | 1 |
| E | Low | Performances | F | Cello | T | T | 6793 | 1 | 1 | 1 |
| F | Low | Documentaries | F | Cap | F | T | 23710 | 1 | 1 | 1 |

○ Aspects in sequence-level data selection criterion  ○ Aspects refined during fine-grained semantic annotation

Figure 3: The representative data of MGIT. Here we illustrate six sequences with different aspects (*e.g.*, narrativity, topics, virtuality, object classes, spatio-temporal continuity, and total frames).

MGIT follows the recently proposed large-scale benchmark VideoCube [19] to conduct the data collection. VideoCube refers to the film narrative (*i.e.*, a chain of causal relationship events occurring in space and time) and proposes the *6D principle* for benchmark construction. In this work, we divide the 6D principle into two parts. Four dimensions (*i.e.*, object class, spatial continuity, temporal continuity, and total frame), together with narrativity and topic, form the new sequence-level selection criterion. The other two dimensions (*i.e.*, motion mode and scene category) will be refined during fine-grained semantic annotation. Therefore, we first regard the original VideoCube as the candidate samples, then add the additional examination of narrativity and topic, and finally select 150 video sequences to form the MGIT. Particularly, the proportions of the *train/val/test* subsets are the same as the original VideoCube. Thus, sequences in each subset are *105/15/30* in MGIT. Taking Figure 3 as an example, here we present several dimensions considered in the data collection process:

**Topic and Narrativity.** We have divided the main video topics into six categories, which are *cartoons*, *movies & TV shows*, *outdoor sports*, *regular sports*, *performances*, and *documentaries*. Among them, cartoons and movies & TV shows usually have a high narrativity (*i.e.*, the video content contains a solid causal relationship, as shown in Figure 3 A and B). Outdoor sports and regular sports contain rich patterns of motion, and these motions can be linked chronologically into a *story*, but the narrativity is usually simple than in cartoons and movies (Figure 3 C and D). Compared to other topics, performances and documentaries usually record one action with low narrativity (Figure 3 E and F). However, these examples are classifications for most cases; it is worth noting that some performances (*e.g.*, sketches with explicitly narrative content on stage) and some documentaries (*e.g.*, documentaries with causal teaching steps) also belong to high narrativity.

**Spatial Continuity and Temporal Continuity.** Temporal continuity means the video content is developed according to the normal time flow (*i.e.*, without fast-forwarding, fast-receding, or interpolation). Spatial continuity means the video content takes place in a fixed space.

**Virtuality.** Virtuality refers that this video is computer-generated, like cartoons or games. The same content in virtuality videos can be very different from videos sampled from the real world; thus, virtuality videos can present a new challenge for object tracking and long video understanding.

## 3.2 Natural Language Annotation

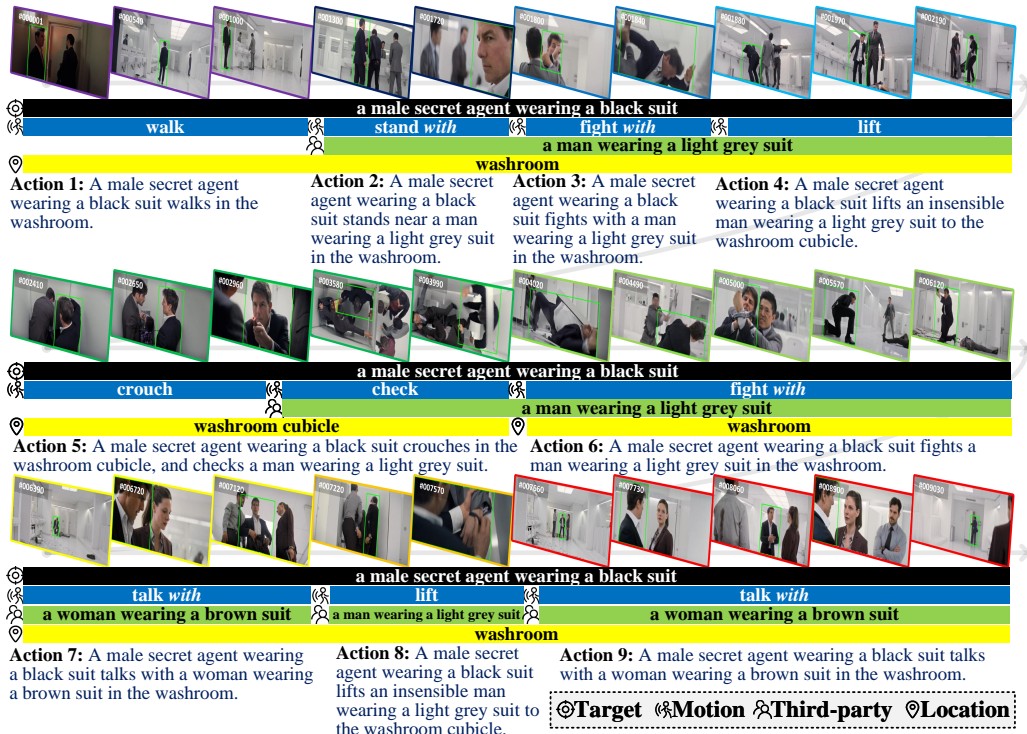

Figure 4: An example of *action* annotation. We label the target, motion pattern, third-party object, and scene for each *action*. The target to be tracked is determined in the first frame and does not change during the entire video sequence. A change in any of the other three elements will end the current *action* and proceed into the following *action*.

In this work, we design a **multi-granular annotation strategy** to provide scientific natural language information. Video content is annotated by three grands (*i.e.*, *action*, *activity*, and *story*, as shown in Figure 1 D1-D3). This hierarchical structure to represent the video content is motivated by existing works in computer vision [20, 43] and human cognitive [21, 22], such as a recent method [43] decouples the video content into multiple granularities for the visual question-answering task [44], aiming to help algorithms better understand video information like humans.

**Action.** As shown in Figure 4, we use the following critical narrative elements to portray an *action*: tracking target (who), motion (what) and third-party object (if present), location (where), and time interval (when). On the one hand, the above elements are necessary to portray narrative content. On the other hand, these elements are also essential grammatical components to form complete sentences. In particular, we use Stanford CoreNLP [45], a widely used natural language processing toolkit, to check the semantic annotations of other multi-modal datasets. We find that more than half of these semantic descriptions are only annotated at the phrase level, lacking the necessary grammatical structure (the detailed statistic result has been shown in Section A.3.1 of the Appendix). Thus, compared with existing works, MGIT can describe more detailed narrative content.

**Activity.** An *action* describes what happens in a short period, while an *activity* can be seen as a collection of *actions* with clear causal relationships. A new *activity* is usually accompanied by a scene switch or an explicit change of the third-party object. Compared with the former *action*, if an *action* is preferred to be the beginning (*i.e.*, reason) of a new event rather than an ending (*i.e.*, result) of an old event, it can be regarded as a starting point of an *activity*. As shown in Figure 4 and Figure 1 (D1-D3), the first four actions describe a complete causality (the target approaches the third party, they fight, and cause the third party to be insensible), while the 5th *action* starts a new event (start examining the unconscious third-party and conduct a second fight when he wakes up). Therefore, the 4th and 5th *actions* can be divided into two different *activities*.

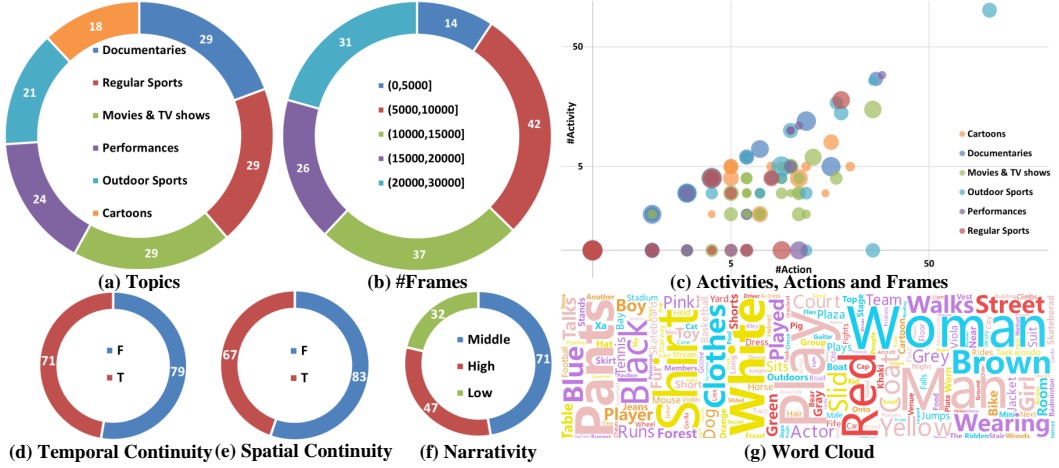

Figure 5: Statistical analysis of key aspects in MGIT. (a-b) Distribution of topics and length of sequences. (c) Distribution of *activities* and *actions*. The bubble diameter is in proportion to the length of a sequence, the vertical coordinate and the horizontal coordinate represent the total *activities* and *actions* of this sequence. (d-f) Distribution of temporal continuity, spatial continuity, and narrativity. (g) The word cloud of semantic descriptions.

**Story.** *Story* is a high-level description. To avoid boring narrativity, we do not stack the existing *actions* and *activities*, but use some words (*e.g.*, first, then, after that, finally, *etc.*) to guide the content, making the temporal and causal more precise.

Based on the data collection process and the multi-granular annotation strategy, we construct MGIT with 2.03 million frames, and provide detailed annotation with 150 *stories*, 621 *activities*, and 982 *actions*. The semantic descriptions contain 77,652 words with 921 non-repetitive words, and more detailed analyses have been illustrated in Figure 5.

## 4 Experimental Results

### 4.1 Datasets and Evaluation Methods

**Datasets.** We select OTB-Lang [1], TNL2k [3], LaSOT [2], and MGIT as experimental environments. Several variants of LaSOT are also concerned: (1) LaSOT_{Ext} [17] is a complement of LaSOT [2] with 150 newly added video sequences. (2) Figure 1 indicates that several semantic descriptions in LaSOT are ambiguous. Thus, 22 ambiguous and 20 unambiguous sequences are selected to

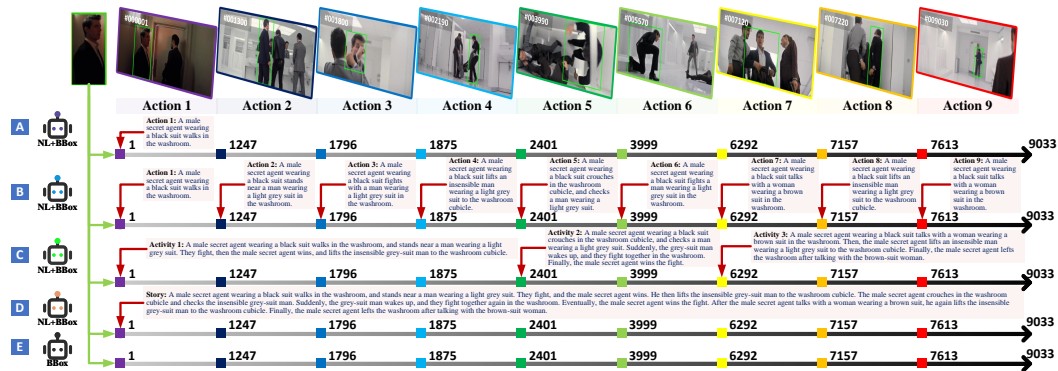

Figure 6: Evaluation mechanisms of visual-based and multi-modal based trackers. (A) Traditional multi-modal tracking mechanism (*i.e.*, only initialize a tracker with BBox and simple semantic information in the first frame). (B-D) Tracking with semantic information updates (*i.e.*, initialize a tracker with BBox and semantic information in the first frame, then update the semantic information in each new interval). (E) Traditional one-pass evaluation (OPE) mechanism (*i.e.*, only initialize a tracker with BBox in the first frame).

form the LaSOT$_{Sub}$, aiming to better analyze tracking performance with different kinds of natural language information. (3) LaSOT$_{NLC}$ is a subset of LaSOT$_{Sub}$, which is formed by the 20 unambiguous sequences, and we have carefully checked all the semantic and visual information in this subset.

**Evaluation Methods.** As shown in Figure 6, various mechanisms are designed to evaluate **tracking precision (PRE)** and **success rate (SR)**. We use $F_t$ to represent the $t$-th frame. (1) Precision is calculated based on the center distance between the predicted BBox $p_t$ and the ground truth BBox $g_t$ (*i.e.*, $d_t = \|c_p - c_g\|_2$, where $c_p$ and $c_g$ represent center points). By calculating the proportion of frames where $d_t \leq \theta_d$ and plotting curves at different thresholds, we can generate a *precision plot*. PRE is common to use $\theta_d = 20$ as the criterion to rank trackers. (2) Furthermore, researchers [19] provide the **normalized precision (N-PRE)** to eliminate the effect of target size. When trackers have a predicted center outside the ground-truth, an additional penalty term, represented by $d_t{}^p$, is included to account for the shortest distance between the center point $c_p$ and the edge of the ground-truth. The final result is then normalized to a range of 0 to 1 (*i.e.*, $N(d_t) = \frac{d_t{}'}{\max(\{d_i{}'|i \in F_t\})}$, where $d_t{}' = d_t + d_t{}^p$). Similarly, the *normalized precision plot* is generated by plotting statistical outcomes derived from various $\theta_d{}'$ values. (3) Besides, frames with the intersection over union (IoU) $\Omega(p_t, g_t) = \frac{p_t \bigcap g_t}{p_t \bigcup g_t} \geq \theta_s$ can be regarded as successful tracking, and the SR measures the percentage of successfully tracked frames. Drawing the results based on various $\theta_s$ is the *success plot*. For more details on the evaluation metrics, please refer to Section B.1 in the Appendix.

Table 1: Results on different multi-modal benchmarks (based on mechanism A in Figure 6).

| Tracker | OTB-Lang [1] | | TNL2k [3] | | LaSOT [2] | | LaSOT$_{Ext}$ [17] | | LaSOT$_{Sub}$ | | LaSOT$_{NLC}$ | | MGIT | |
|---|---|---|---|---|---|---|---|---|---|---|---|---|---|---|
| | PRE | SR | PRE | SR | PRE | SR | PRE | SR | PRE | SR | PRE | SR | PRE | SR |
| SNLT [46] | 0.848 | 0.666 | 0.081 | 0.100 | 0.475 | 0.459 | 0.306 | 0.262 | 0.527 | 0.495 | 0.513 | 0.483 | 0.004 | 0.036 |
| VLT_SCAR [42] | 0.898 | 0.739 | 0.556 | 0.497 | 0.677 | 0.630 | 0.503 | 0.428 | 0.670 | 0.633 | 0.659 | 0.633 | 0.124 | 0.177 |
| VLT_TT [42] | 0.931 | 0.764 | 0.583 | 0.539 | 0.714 | 0.670 | 0.549 | 0.465 | 0.707 | 0.660 | 0.721 | 0.662 | 0.324 | 0.474 |
| JointNLT [18] | 0.856 | 0.653 | 0.598 | 0.552 | 0.640 | 0.607 | 0.457 | 0.398 | 0.624 | 0.583 | 0.707 | 0.651 | 0.433 | 0.603 |

## 4.2 Comparison with Other Multi-modal Benchmarks (Mechanism A)

We select several SOTA multi-modal trackers as baseline models and evaluate them on various benchmarks (as shown in Table 1). To fairly compare the tracking performance on MGIT and other datasets, we only allow trackers to use the semantic information of the first *action* in this experiment. Results show that: (1) most trackers perform worst on MGIT, which means it is a more complex environment with more challenges. (2) By comparing the tracking results on LaSOT$_{Sub}$ and LaSOT$_{NLC}$, we can find that most trackers perform worse on LaSOT$_{Sub}$, showing that ambiguous

semantic information may introduce external interferences. Thus, we avoid this problem via the scientific annotation and check process for MGIT construction.

## 4.3 Experimental Results on MGIT

**Tracking by NL&BBox (Mechanism B-D).** As shown in Figure 6 (B-D), both visual information (BBox of the first frame) and semantic information (natural language description) can be used for multi-modal trackers. Specifically, different granularities have various lengths, while most trackers have a maximum limit of the input semantic information. JointNLT [18] sets 50 as a maximum limit and truncates the excess information. This truncation occurs for both *activity* (C) and *story* (D). Similarly, the VLT [42] series limits the semantic length but can avoid truncation by adjusting the parameters. Thus, we set the semantic length to 80 for the *activity* and 200 for the *story*, with zero padding as necessary. From Table 2, we can draw the following conclusions: (1) SNLT, VLT_SCAR, and VLT_TT perform well when using longer semantic information like *activity* and *story*. This indicates that the semantic information processing modules (BERT [47]) used in these trackers can effectively handle long text. Besides, their good performances in *activity* indicate that as an intermediate granularity, *activity* accomplishes a balance between the amount of information and the number of semantic description updates. (2) On the contrary, JointNLT performs well on *action* rather than levels with longer descriptions, suggesting that truncated semantic information leads to poorer performance. Therefore, to obtain better multi-modal information processing capabilities, algorithms should first ensure that long texts can be processed rather than truncated directly.

Table 2: Results of different trackers on MGIT.

| Tracker | Architecture | Initialize | Mechanism | PRE | N-PRE | SR |
|---|---|---|---|---|---|---|
| SiamCAR [11] | SNN | BBox | | 0.116 | 0.378 | 0.183 |
| SiamRCNN [10] | SNN | BBox | | 0.512 | 0.707 | 0.591 |
| PrDiMP [12] | SNN+CF | BBox | | 0.296 | 0.602 | 0.453 |
| KeepTrack [13] | SNN+CF | BBox | E | 0.373 | 0.695 | 0.519 |
| TransT [39] | Transformer | BBox | | 0.447 | 0.670 | 0.539 |
| MixFormer [14] | Transformer | BBox | | 0.526 | 0.775 | 0.629 |
| OSTrack [15] | Transformer | BBox | | 0.476 | 0.706 | 0.583 |
| GRM [16] | Transformer | BBox | | 0.500 | 0.718 | 0.597 |
| SNLT [46] | SNN | NL&BBox | Action (B) | 0.004 | 0.226 | 0.036 |
| | | | Activity (C) | 0.004 | 0.234 | 0.038 |
| | | | Story (D) | 0.005 | 0.230 | 0.040 |
| VLT_SCAR [42] | SNN | NL&BBox | Action (B) | 0.116 | 0.354 | 0.167 |
| | | | Activity (C) | 0.124 | 0.382 | 0.180 |
| | | | Story (D) | 0.127 | 0.403 | 0.184 |
| VLT_TT [42] | Transformer | NL&BBox | Action (B) | 0.318 | 0.602 | 0.468 |
| | | | Activity (C) | 0.325 | 0.627 | 0.485 |
| | | | Story (D) | 0.322 | 0.616 | 0.480 |
| JointNLT [18] | Transformer | NL&BBox | Action (B) | 0.445 | 0.786 | 0.610 |
| | | | Activity (C) | 0.441 | 0.780 | 0.605 |
| | | | Story (D) | 0.433 | 0.773 | 0.600 |

By comparing results under mechanisms A and D, we can find that in this complex environment, well-designed trackers (*i.e.*, trackers with suitable long input processing ability) can perform better via longer descriptions than only relaying a short description (SNLT: 0.036 → 0.040, VLT_SCAR: 0.177 → 0.184, VLT_TT: 0.474 → 0.480 in SR). The above experiments indicate two key points: (1) Richer semantic information (mechanism D based on *story*) can improve the tracking performance than a simple sentence (mechanism A based on information for the first *action*), which can also verify the accuracy and necessity of the proposed multi-granularity semantic annotation strategy. (2) Only providing a simple description for multi-modal trackers is unreasonable for MGIT. Thus, initializing the tracking process with longer and more specific sentences, or updating the semantic information periodically throughout the sequence, has been found to be more effective in accurately locating targets within complex scenes.

**Tracking by BBox Only (Mechanism E).** We mainly evaluate SOTA visual-based trackers under mechanism E. As shown in Table 2, by comparing with other trackers, the transformers-based trackers have emerged as the predominant approach and achieved SOTA performance. Besides, it is worth noting that visual-based trackers usually outperform multi-modal trackers. Although we hope that more modal information can improve the tracking performance, the current multi-modal approaches cannot better align different modalities, resulting in the multi-modal information not being fully utilized. In contrast, visual-based methods have been well developed over the past decades and can better use visual features to accomplish good tracking performance. This result (*i.e.*, current multi-modal trackers are worse than visual-based trackers) can also be found in other works, highlighting the significant room for improvement in multi-modal tracking. More detailed experimental results and analyses can be found in Section B.3 of the Appendix.

## 4.4 Visualization and Bad Case Analysis

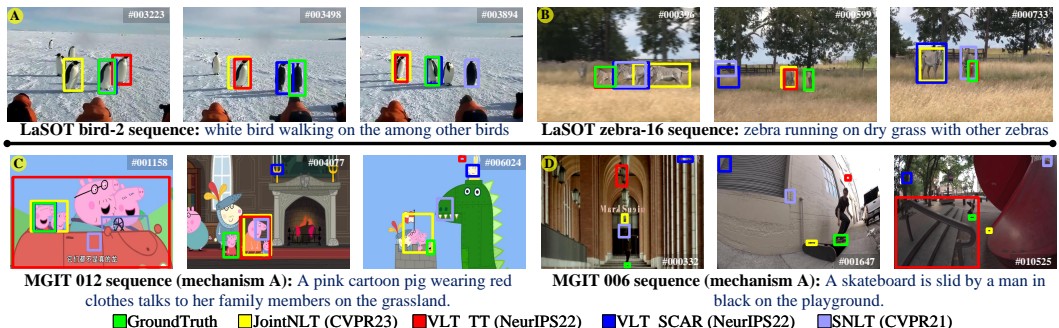

**LaSOT bird-2 sequence:** white bird walking on the among other birds

**LaSOT zebra-16 sequence:** zebra running on dry grass with other zebras

**MGIT 012 sequence (mechanism A):** A pink cartoon pig wearing red clothes talks to her family members on the grassland.

**MGIT 006 sequence (mechanism A):** A skateboard is slid by a man in black on the playground.

■ GroundTruth ■ JointNLT (CVPR23) ■ VLT_TT (NeurIPS22) ■ VLT_SCAR (NeurIPS22) ■ SNLT (CVPR21)

Figure 7: Bad cases of representative multi-modal trackers on LaSOT [2] and MGIT. (A-B) Ambiguous semantic annotations on LaSOT lead trackers to locate at similar objects. (C-D) The mechanism A used in existing multi-modal SOT benchmarks is unable to adapt to complex scenes like MGIT.

We further analyze the bottlenecks of the multi-modal algorithms through the bad cases shown in Figure 7. The first two examples are selected from LaSOT [2], demonstrating that ambiguous semantic information may introduce noise, leading algorithms to wrongly focus on similar objects – this emphasizes the importance of accurate semantic annotations. The latter two examples are chosen from MGIT, demonstrating that the experimental environment constructed by MGIT presents complex spatio-temporal and causal relationships, posing challenges to multi-modal algorithms. Specifically, the complexity of MGIT results in significant differences between the target appearance and background environment in the initial frame and subsequent states. Besides, MGIT is selected from the recently released VideoCube [19] benchmark, which has a higher image resolution, posing challenges for trackers to relocate the target after failure. Additionally, using only the first action information (mechanism A) is applied in all other multi-modal SOT benchmarks. However, it is not applicable to visual object tracking in complex scenes like MGIT (Figure 7(C-D)). Therefore, the proposed multi-granularity annotation strategy offers a more reasonable solution. Multi-modal trackers who want to perform better on MGIT need a more well-designed semantic information processing module to accurately extract useful information described by semantic labels. Nevertheless, existing trackers have not made specialized designs for this aspect, which can be further improved.

## 5 Conclusions

**Summary.** Accuracy target tracking is the foundation for accomplishing high-level tasks like long video understanding, and introducing natural language into visual object tracking is a possible way to increase tracking ability. Different from existing multi-modal benchmarks that mainly consisted of short sequences with simple or even ambiguous descriptions, we (1) propose a new multi-modal benchmark named **MGIT** with 150 long video sequences, and (2) design a **multi-granular annotation strategy** for generating scientific semantic information. On the one hand, MGIT is a challenging and complex environment for visual tracking and video understanding research (*i.e.*, trackers should process the spatio-temporal and causal relationships coupled with longer narrative content to accomplish better performance). On the other hand, the multi-granular annotation strategy models the human cognitive enhancement process, which may provide a step-by-step "learning" environment for generating human-like trackers. The experimental results demonstrate that MGIT is a more complex environment, and our proposed strategy is a feasible solution for coupling human understanding into semantic labels. Besides, existing trackers still have a large room for development, like improving the capability for processing long text and aligning multi-modal information. Conclusionally, we hope this work can help researchers to conduct further research in object tracking and video understanding.

**Limitations.** Some limitations here can be further enhanced in future work. First, we can expand MGIT with more types of videos to provide a more complicated environment for data-driven algorithms. Besides, we can design a better comprehensive evaluation system to measure visual tracking and video understanding ability. Finally, we can add more types of tasks based on the benchmark, and try to test algorithms for tasks like video caption and action recognition.

## Acknowledgments and Disclosure of Funding

This work was supported in part by the National Key R&D Program of China (No.2022ZD0116403); the National Natural Science Foundation of China (No.61721004); the Strategic Priority Research Program of Chinese Academy of Sciences (No.XDA27000000).

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

# Appendix

## A   Dataset Information

### A.1   Basic Information

In this work, we propose a new **m**ulti-modal **g**lobal **i**nstance **t**racking benchmark named **MGIT**. It consists of 150 long video sequences with a total of 2.03 million frames, aiming to fully represent the complex spatio-temporal and causal relationships coupled in longer narrative content.

This work further expands our former work, which was accepted by IEEE TPAMI in 2022. We proposed a global instance tracking task in the previous work and released an online evaluation platform (URL: http://videocube.aitestunion.com). We hope the online platform can help researchers use our proposed resources better and conduct more fair comparisons via our real-time evaluation server.

In the past year, our platform has evaluated 380 algorithms and received more than 287k IP visits from 130 countries (statistics by Jan 04, 2024). However, all submitted algorithms do not show significant performance improvement on the GIT task, and their tracking performances are significantly degraded on the GIT task compared to other representative single object tracking benchmarks (*i.e.*, short-time tracking and long-time tracking). This phenomenon shows the limitations of a single visual modality for long video understanding of complex narrative relationships. Thus, we conduct this work to introduce semantic information, which aims to help the algorithms better cope with the challenges posed by complex narrative relationships for target tracking and long video understanding.

Since the motivation for this work is closely inherited from our former work, and the existing online platform has received considerable attention worldwide, we select to release the MGIT benchmark via the same online platform. The proposed benchmark, experimental results, and toolkit will be released gradually on http://videocube.aitestunion.com/ (Figure A1).

Our dataset has been uploaded to OneDrive and Baiduyun disk, and the online evaluation platform is maintained by dedicated staff, which will ensure the stability of the dataset.

We declare that we bear all responsibility in case of violation of rights, *etc.*, and confirm the data license. Our work is licensed under CC BY-NC-SA 4.0. Users are free to use the dataset for research purposes.

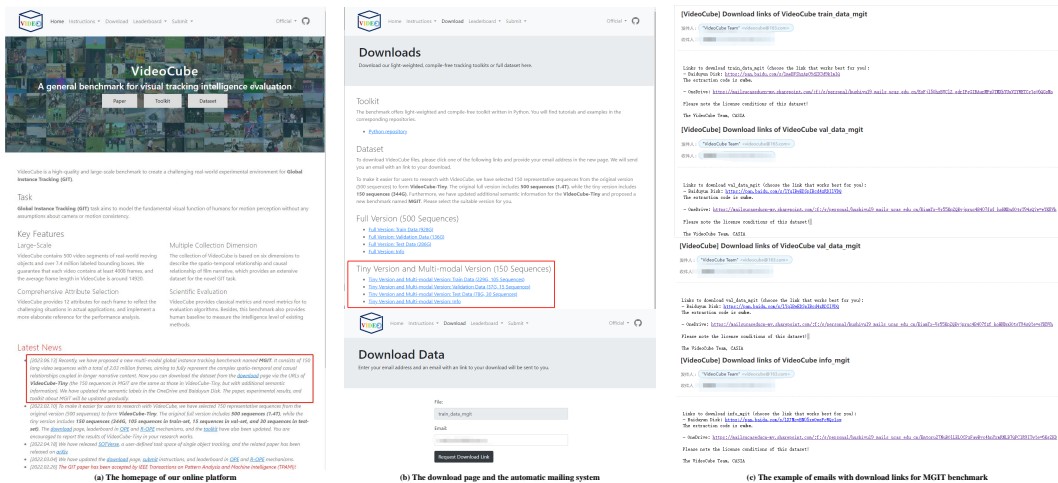

Figure A1: Our online platform and currently updated download links with related instructions.

### A.2   Data Selection

The 150 sequences in MGIT are carefully selected from the original VideoCube [19] benchmark. In the video selecting part, we thoroughly consider the consistency between the new dataset (MGIT)

and the original dataset (VideoCube) in various dimensions (6D principle), while also taking into account the consistency of difficulty.

The specific process is as follows. (1) We assess the similarity in the distribution of the selected MGIT dataset to the original VideoCube dataset across various dimensions, including object class, scene category, motion mode, and more, in accordance with the 6D principle. (2) Besides, it is essential for the selected dataset to maintain a similar level of difficulty as the original VideoCube dataset. Regarding the difficulty level, we select three state-of-the-art trackers (MixFormer [14], KeepTrack [13], and SiamRCNN [10]) with different architectures as the basis for our selection criteria. The success scores (based on IoU) of these algorithms on the original 500 sequences are calculated and ranked to measure sequence difficulty. (3) By considering both the distribution across the 6D principle and the difficulty level, we carefully choose 150 representative sequences to construct the MGIT dataset, aiming to maintain consistency with the distribution of the original VideoCube dataset.

## A.3 Semantic Annotation

### A.3.1 Semantic Annotation Deficiencies of Existing Multi-modal SOT Benchmarks

To better show the semantic annotation deficiencies of existing multi-modal SOT benchmarks, we conduct statistical analyses on OTB-Lang [1], LaSOT [2], LaSOT$_{Ext}$ [17], and TNL2k [3] from two aspects:

1. **Ambiguity of semantic labeling:** A random sampling inspection is conducted to address the ambiguity of semantic annotation. Specifically, 10 sequences are randomly selected from each dataset for inspection. To ensure random and fair selection, all sequences in each dataset are alphabetically sorted first, and samples are taken at equal intervals.

2. **Completeness of grammatical structures:** Semantic descriptions of high quality typically necessitate complete grammatical structures. Hence, Stanford CoreNLP [45] is utilized to analyze the semantic labels in all four datasets, and statistics those that adhered to the criteria of complete sentences.

Table A3: The statistics of semantic annotation quality in four representative datasets.

| Benchmark | OTB-Lang [1] | LaSOT [2] | LaSOT$_{Ext}$ [17] | TNL2k [3] |
|---|---|---|---|---|
| **Statistical Analysis 1:** Inspection Pass Rate (Non-ambiguous Semantic Descriptions) | 30% | 70% | 60% | 60% |
| **Statistical Analysis 2:** Complete Sentences Rate (Checked by Stanford CoreNLP, Including Complete Grammatical Structures) | 9% | 63% | 36% | 20% |

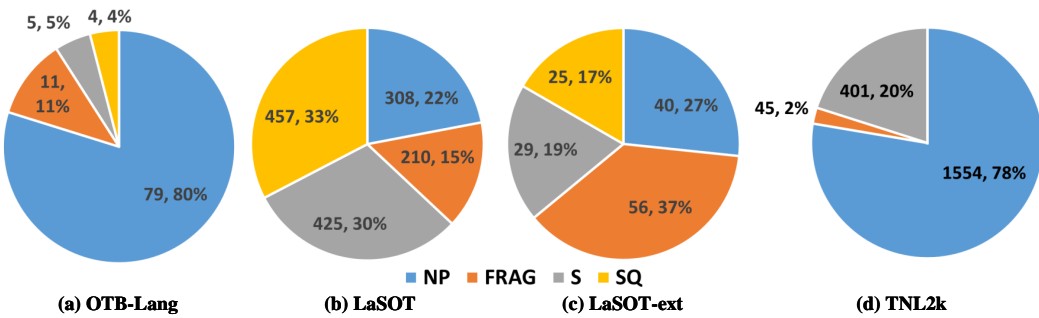

(a) OTB-Lang          (b) LaSOT          (c) LaSOT-ext          (d) TNL2k

Figure A2: Statistical analysis about the completeness of grammatical structures, based on Stanford CoreNLP [45] toolkit. (NP: noun phrase; FRAG: fragment; S: simple declarative clause; SQ: inverted yes/no question, or main clause of a wh-question. Only S and SQ satisfy the completeness of the grammatical structure.)

As shown in Table A3 and Figure A2, the current datasets exhibit deficiencies regarding ambiguity and completeness. However, our proposed MGIT benchmark considers these factors during construction and avoids the previously mentioned issues, thus possessing higher-quality semantic annotations.

### A.3.2 Annotation Process

We chose an elite annotation team instead of crowdsourcing to carry out this work and ensured quality through multiple efforts.

1. **Task Decomposition.** We first decompose the task to ensure a standardized workflow for execution. For instance, we begin by annotating the finest granularity (action), and subsequently continue with the annotations of activity and story. This approach ensures accuracy and consistency in the fundamental content throughout various levels of granularity. Given that action is the finest granularity and its annotation quality may affect activity and story, we refer to film narrative literature and English grammar materials to further decompose the description of an action. This decomposition involves identifying the tracking target (who), the motion (what), the presence of a third-party object (if applicable), the location (where), and the time interval (when). By obtaining these specific details, annotators can attain a standardized and comprehensive description of the action.

2. **Annotator Selection.** Considering the difficulty of controlling annotation quality in crowdsourcing, we chose highly cognitive graduate students with experience in dataset annotation to form an annotation team. Team members not only have experience in annotating vision-based datasets represented by VideoCube but also have experience in annotating image datasets in visual psychology. They have a solid foundation in dataset construction in fields such as computer vision and cognitive psychology. All team members undergo standardized training before formal annotation to ensure their understanding of task characteristics and annotation rules. Additionally, the training session includes 10 video examples of different types, requiring annotators to comprehend the annotation process and details.

3. **Annotation Workflow.** (1) The formal annotation process begins, wherein annotation personnel is grouped based on video types, including cartoons, movies, TV shows, sports, performances, and documentaries. Any issues requiring discussion will be documented, followed by a comprehensive discussion among all personnel, and then the annotation process for that particular sequence will commence. (2) For instance, in the case of a sniper rifle as the target, which term should be employed: "gun" or "sniper rifle"? Through consulting relevant materials and conducting discussions, we have concluded that the fundamental principle of annotation is to incorporate human comprehension into semantic labels. As there is no second firearm in the sequence and the term "gun" encompasses "sniper rifle" semantically, the usage of "gun" aligns with commonplace terminology. Nevertheless, if a second gun appears in the sequence, "sniper rifle" may be employed to underscore the target's distinctiveness. (3) Furthermore, to enhance the standardization of annotations, we refer to WordNet to construct verb and noun lists. Initially, annotators will choose candidate terms from the current vocabulary lists to depict the essential elements of the scene, aiming to maintain consistency in the portrayal of actions across varying sequences to the greatest extent possible. If there are no appropriate terms found in the candidate list, annotators will employ new vocabulary to depict the elements and subsequently incorporate them into the candidate list, supplemented with relevant examples for future annotation reference.

4. **Quality Review.** After completing the annotation for all sequences, we will review the content to ensure its quality. Additionally, we utilize the Stanford CoreNLP [45], a natural language processing toolkit, to examine the semantic annotations and ensure the grammatical structure's completeness.

### A.3.3 Annotation File

We propose a **multi-granular annotation strategy** to generate the semantic description, and use JSON format to save the natural language annotation for each video sequence. Here we illustrate an example to show the JSON file structure for video sequence *001* in the MGIT benchmark, as shown in Listing 1. Due to the limited space, we only illustrate some representative information, while the remaining information with similar structure is indicated by ellipses. Please download and check the dataset for more detailed annotation about each video sequence.

```json
{
    "action": {"action_1": {
            "start_frame": 0,
            "end_frame": 1246,
            "length": 1247,
            "object_class": "male secret agent",
            "appearance": "black suit",
            "action_1": "walk",
            "prep_1": NaN,
            "3rd_object_1": NaN,
            "action_2": NaN,
            "prep_2": NaN,
            "3rd_object_2": NaN,
            "scene": "washroom",
            "description": "A male secret agent wearing a black suit
            ↪  walks in the washroom"},
        "action_2": {...},
        "action_3": {...},
        "action_4": {...},
        "action_5": {...},
        "action_6": {...},
        "action_7": {...},
        "action_8": {...},
        "action_9": {...},},
    "activity": {"activity_1": {
            "start_frame": 0,
            "end_frame": 2400,
            "length": 2401,
            "description": "A male secret agent wearing a black suit
            ↪  walks in the washroom, and stands near a man wearing
            ↪  a light grey suit. They fight, then the male secret
            ↪  agent wins, and lifts the insensible grey-suit man
            ↪  to the washroom cubicle."},
        "activity_2": {...},
        "activity_3": {...},},
    "story": {"story_1": {
            "start_frame": 0,
            "end_frame": 9032,
            "length": 9033,
            "description": "A male secret agent wearing a black suit
            ↪  walks in the washroom, and stands near a man wearing
            ↪  a light grey suit. They fight, and the male secret
            ↪  agent wins. He then lifts the insensible grey-suit
            ↪  man to the washroom cubicle. The male secret agent
            ↪  crouches in the washroom cubicle and checks the
            ↪  insensible grey-suit man. Suddenly, the grey-suit
            ↪  man wakes up, and they fight together again in the
            ↪  washroom. Eventually, the male secret agent wins the
            ↪  fight. After the male secret agent talks with a
            ↪  woman wearing a brown suit, he again lifts the
            ↪  insensible grey-suit man to the washroom cubicle.
            ↪  Finally, the male secret agent lefts the washroom
            ↪  after talking with the brown-suit woman."}}}
```

Listing 1: The JSON file about the semantic information of video sequence *001*.

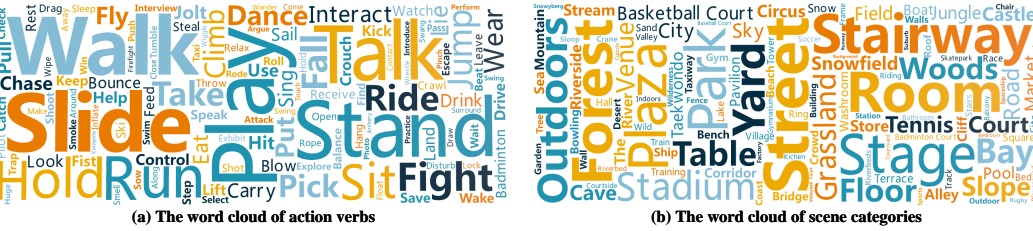

(a) The word cloud of action verbs    (b) The word cloud of scene categories

Figure A3: The word cloud of action verbs and scene categories on MGIT.

1. **Action**: For each *action*, we save the following information in the JSON file:

   (a) *start_frame*: The starting frame of the *action*. Note that the original VideoCube is labeled with the first frame starting from 0. Therefore, in the JSON file, we use 0 to represent the first frame (Note that in the figures of the main paper, we use the same format as the other datasets to show the starting point as 1 for ease of understanding).

   (b) *end_frame*: The ending frame of the *action*.

   (c) *length*: Length of the time interval.

   (d) *object_class*: The object class of the tracking target.

   (e) *appearance*: The appearance of the tracking target. We ensure that the description of the target's appearance is unique in the entire sequence.

   (f) *action_1*: The first action of the target. The word cloud of action verbs is illustrated in Figure A3 (a).

   (g) *prep_1*: The preposition of action (if present).

   (h) *3rd_object_1*: The interaction object of the first action (if present).

   (i) *action_2*: The second action of the target (if present).

   (j) *prep_2*: The preposition of action (if present).

   (k) *3rd_object_2*: The interaction object of the second action (if present.

   (l) *scene*: The scene category. The word cloud of action verbs is illustrated in Figure A3 (b).

   (m) *description*: The natural language description.

2. **Activity**: For each *activity*, we save the following information in the JSON file:

   (a) *start_frame*: The starting frame of the *activity*.

   (b) *end_frame*: The ending frame of the *activity*.

   (c) *length*: Length of the time interval.

   (d) *description*: The natural language description.

3. **Story**: For each *story*, we save the following information in the JSON file:

   (a) *start_frame*: The starting frame of the *story*.

   (b) *end_frame*: The ending frame of the *story*.

   (c) *length*: Length of the time interval.

   (d) *description*: The natural language description.

## A.4 Dataset Structure

The MGIT benchmark includes 150 long video sequences (344G) with detailed annotations. We add semantic information based on a multi-granularity annotation strategy while retaining the detailed annotation information of the original VideoCube dataset, aiming to help multi-modal methods better understand the narrative content of long videos.

The dataset download and file organization process is as follows:

1. Download three subsets (*train/val/test*) and the info data. Please click on the hyperlink to visit our dataset (choose the link that works best for you).

   (a) **Train Data (229G, 105 Sequences)**:

```
1   |-- val/
2       |   |-- 005/
3       |   |   |-- frame_005/
4       |   |   |   |-- 000000.jpg/
5       |   |   |         ......
6       |   |   |   |-- 016891.jpg/
7       |   |-- 029/
8       |   |      ......
9       |   |-- 362/
```

Listing 2: The dataset structure of *val* subset.

```
1   |-- MGIT/
2       |   |-- data/
3       |   |   |-- train/
4       |   |   |   |-- 002/
5       |   |   |   |      ......
6       |   |   |   |-- 480/
7       |   |   |-- val/
8       |   |   |   |-- 005/
9       |   |   |   |      ......
10      |   |   |   |-- 362/
11      |   |   |-- test/
12      |   |   |   |-- 001/
13      |   |   |   |      ......
14      |   |   |   |-- 498/
15      |   |   |-- train_list.txt
16      |   |   |-- val_list.txt
17      |   |   |-- test_list.txt
18      |   |-- attribute/
19      |   |   |-- absent/
20      |   |   |-- color_constancy_tran/
21      |   |   |      ......
22      |   |   |-- description/
23      |   |   |      ......
24      |   |   |-- shotcut/
```

Listing 3: The dataset structure of the MGIT benchmark.

      i. **OneDrive**
     ii. **Baiduyu Disk** (The extraction code is **cube**.)
  (b) **Validation Data (37G, 15 Sequences)**:
      i. **OneDrive**
     ii. **Baiduyu Disk** (The extraction code is **cube**.)
  (c) **Test Data (78G, 30 Sequences)**:
      i. **OneDrive**
     ii. **Baiduyu Disk** (The extraction code is **cube**.)
  (d) **Info Data (89.16M, 15 Attributes)**:
      i. **OneDrive**
     ii. **Baiduyu Disk** (The extraction code is **cube**.)

2. Check the number of files in each subset and run the unzipping script. To facilitate transmission and downloading, the very long video sequences in the dataset are divided into smaller segments during the packaging process. Each segment is compressed and kept under

4GB. For instance, in the train set, the sequence 013 is divided into three compressed files: frame_013_split.z01, frame_013_split.z02, and frame_013_split.zip. Before unzipping:

  (a) The *train* subset should include 129 files (128 data files and an unzip_train bash).

  (b) The *val* subset should include 22 files (21 data files and an unzip_val bash).

  (c) The *test* subset should include 41 files (40 data files and an unzip_test bash).

3. Run the unzipping script in each subset folder, and delete the script after decompression.

4. Taking *val* subset of full version as an example, the folder structure is listed as Listing 2.

5. Unzip attribute.zip in info data. Attention that we only provide properties files for *train* and *val* subsets. For ground-truth files in the test subset, we only offer a small number of annotations for restart frames to support the essential function of the R-OPE mechanism (For detailed information about the R-OPE mechanism, please refer to the TPAMI paper [19] about GIT task and the VideoCube benchmark. Note that we only use the OPE mechanism for MGIT evaluation process, while the R-OPE mechanism is supported for visual-based trackers.). The annotations of other frames in the test subset have been set as zero. Please upload the final results to the server (http://videocube.aitestunion.com/) for evaluation.

6. Rename and organize folders as Listing 3. Note that the semantic information (saved as JSON file) for the MGIT benchmark is saved in the **description** folder.

## B  Experiment Information

### B.1  Evaluation Metrics

Assume an experiment dataset $E$ (*e.g.*, MGIT) comprises $|E|$ sequences, with $|\cdot|$ representing the cardinality. In the sequence $L$, we use $F_t$ to represent the $t$-th frame. We assume that $p_t$ denotes the predicted position by the tracker $T$, and $g_t$ refers to the ground-truth position. Notably, if a frame does not contain the target (*i.e.*, full-occlusion or out-of-view), it is considered an empty set (i.e., $g_t = \phi$) and thereby excluded from the evaluation process. The *precision score* and *success score* of frame $F_t$ are calculated through the following formulas:

$$
\begin{aligned}
d_t &= \|c_p - c_g\|_2, \\
s_t &= \Omega(p_t, g_t) = \frac{p_t \bigcap g_t}{p_t \bigcup g_t},
\end{aligned}
\tag{1}
$$

where $d_t$ represents the distance between the center points $c_p$ and $c_g$, while $\Omega(\cdot)$ denotes the intersection over union.

Recently, the *normalized precision score* ([19]) is proposed to eliminate the impact of the target size and frame resolution. In the case where trackers have a predicted center outside the ground-truth, an additional penalty item $d_t^{\,p}$ is included, representing the shortest distance between the center point $c_p$ and the edge of the ground-truth. If the center point of a tracker falls within the ground-truth, the center distance $d_t^{\,'}$ is equal to the original precision $d_t$, resulting in $d_t^{\,p} = 0$:

$$
\begin{aligned}
N(d_t) &= \frac{d_t^{\,'}}{\max(\{d_i^{\,'} \mid i \in F_t\})}, \\
d_t^{\,'} &= d_t + d_t^{\,p}.
\end{aligned}
\tag{2}
$$

The precision $P(E)$, normalized precision $N(E)$, and success $S(E)$ of environment $E$ are defined as follows:

$$P(E) = \frac{1}{|E|} \sum_{l=1}^{|E|} \frac{1}{|L|} |\{t : d_t \leq \theta_d\}|,$$

$$N(E) = \frac{1}{|E|} \sum_{l=1}^{|E|} \frac{1}{|L|} |\{t : N(d_t) \leq \theta_d'\}|, \qquad (3)$$

$$S(E) = \frac{1}{|E|} \sum_{l=1}^{|E|} \frac{1}{|L|} |\{t : s_t \geq \theta_s\}|.$$

The *precision plot* is generated by calculating the proportion of frames with a distance $d_t$ less than or equal to $\theta_d$ and plotting the statistical results across different $\theta_d$ values. In most cases, existing benchmarks use $\theta_d = 20$ as a standard threshold to rank trackers.

The *normalized precision plot* is generated similarly by plotting statistical results obtained from varying $\theta_d'$ values within the range of [0,1]. However, directly assigning a specific $\theta_d'$ value to rank trackers can introduce subjective biases. Therefore, the ranking of trackers is based on the proportion of frames in which the predicted center $c_p$ successfully falls within the ground-truth rectangle $g_t$.

The *success plot* is generated by plotting the results obtained from different overlap thresholds, $\theta_s$, on a curve. In this plot, the mAO (mean average overlap) is commonly utilized to rank trackers.

### B.2 Baseline Information

All experiments are performed on a server with 4 NVIDIA TITAN RTX GPUs and a 64 Intel(R) Xeon(R) Gold 5218 CPU @ 2.30GHz. Detailed information about the baselines are illustrated in Table A4, we use the parameters provided by the original authors.

Table A4: Table: The model architectures and URLs of open-sourced algorithms used in this work.

| Tracker | Architecture | Initializa | URL |
|---|---|---|---|
| SiamCAR [11] | SNN | BBox | https://github.com/ohhhyeahhh/SiamCAR |
| SiamRCNN [10] | SNN | BBox | https://github.com/VisualComputingInstitute/SiamR-CNN |
| PrDiMP [12] | SNN | BBox | https://github.com/visionml/pytracking |
| KeepTrack [13] | SNN | BBox | https://github.com/visionml/pytracking |
| TransT [39] | Transformer | BBox | https://github.com/chenxin-dlut/TransT |
| MixFormer [14] | Transformer | BBox | https://github.com/MCG-NJU/MixFormer |
| OSTrack [15] | Transformer | BBox | https://github.com/botaoye/OSTrack |
| GRM [16] | Transformer | BBox | https://github.com/Little-Podi/GRM |
| SNLT [46] | SNN | NL&BBox | https://github.com/fredfung007/snlt |
| VLT_SCAR [42] | SNN | NL&BBox | https://github.com/JudasDie/SOTS |
| VLT_TT [42] | Transformer | NL&BBox | https://github.com/JudasDie/SOTS |
| JointNLT [18] | Transformer | NL&BBox | https://github.com/lizhou-cs/JointNLT |

*Note: SNN-Siamese Neural Network. NL-Natural Language. BBox-Bounding Box.*

### B.3 More Detailed Experimental Results on MGIT

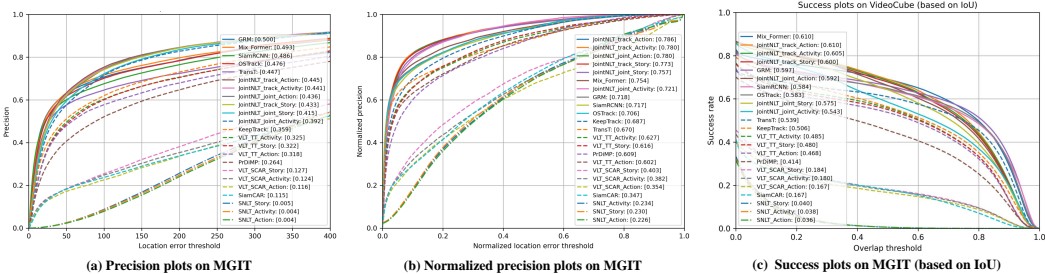

(a) Precision plots on MGIT     (b) Normalized precision plots on MGIT     (c) Success plots on MGIT (based on IoU)

Figure A4: The precision plot (a), normalized precision plot (b), and the success plot (c) on MGIT.

Figure A4 illustrates the precision plot (a), normalized precision plot (b), and the success plot (c). The performance of trackers indicates that the multi-modal trackers still exhibits a certain gap when compared to visual-based trackers.

Specially, the multi-modal tracker SNLT [46] performs poorly in MGIT. The possible reasons behind the poor results of SNLT are as follows: (1) SNLT is based on the local search, which exhibits a performance gap to the global search trackers. Experiments reveal that local search trackers may encounter a more severe tracking drift problem in the MGIT task (this method tracks by cutting out the search area from the original image, while the high image resolution in MGIT will challenge it). Besides, SNLT's weaker tracking ability is more prone to failure. The errors generated as a result will further misguide, thus creating a negative loop. For example, when it loses the target or drifts towards a similar object, it will persist in tracking failure until the target reemerges within its search area with more recognizable visual information. (2) In addition, there are some issues within SNLT's open-source code, and although we fix them during our reproduction and evaluation, SNLT has some limitations in terms of project completeness compared to other better-maintained multi-modal open-sourced trackers, which may also contribute to its poor performance.

Besides, some other challenges may also influence the tracking performance. Here, we discuss the challenges trackers face from two perspectives: by comparing different multi-modal information provisioning mechanisms, and from limitations between single-modal and multi-modal approaches.

1. **Comparison of different multi-modal evaluation mechanisms:** (1) First, compared to mechanism A, which only provides semantic information for the first action, multi-modal methods do not show a significant performance decrease in mechanisms B, C, and D. Except JointNLT [18], the success rate scores of other multi-modal trackers in mechanism D are superior to mechanism A. (2) Furthermore, most algorithms performed the worst in mechanism A because they only obtained semantic descriptions for the first action, and this semantic information is not updated in the subsequent process, which may introduce noise to the tracker as the sequence progresses. (3) Mechanisms B and C both regularly update the semantic information during tracking. However, C has a moderate frequency of semantic information updates, and each update provides a moderate length of semantic information, which may better leverage the capabilities of trackers. (4) Mechanism D, similar to mechanism A, only provides semantic information in the initial frame but offers story information that can cover the entire video. However, current multi-modal trackers lack well-designed semantic understanding modules for handling long texts, making it challenging to align semantic information with visual information

2. **Comparison between single-modal and multi-modal approaches:** Semantic modalities can provide information beyond superficial features such as appearance and location compared to pure visual trackers. However, achieving better correlation and fusion between modalities still needs to be solved. Previous research on other multi-modal tasks has experimentally and theoretically demonstrated that multi-modal approaches can introduce more information, resulting in improved algorithms [48]. However, in the field of SOT research, the performance of multi-modal algorithms still lags behind that of single-modal algorithms. The main reason may be related to the need for more high-quality benchmarks – existing multi-modal benchmarks have significant limitations in the completeness of semantic information and video complexity, making it challenging to provide a favorable experimental environment for multi-modal trackers. Additionally, these multi-modal benchmarks do not adopt a multi-granular annotation strategy, resulting in an evaluation system that only involves mechanism A. As a result, they cannot thoroughly explore the current methods' bottlenecks like our work.

Therefore, the emergence of MGIT can provide a high-quality experimental environment for research, and help the algorithms quickly identify bottleneck issues under various evaluation mechanisms, thereby accelerating the development of efficient multi-modal trackers.

Furthermore, the multi-modal trackers demonstrate superior performance in terms of the normalized precision plot (Figure A4 (b)). We attribute this to the integration of the semantic modality, which enables the multi-modal tracker to effectively recognize the target's position. This observation aligns with intuition, but there is room for further improvement in localization accuracy.

## B.4    A Comparison Experiment about Using Various Granularities

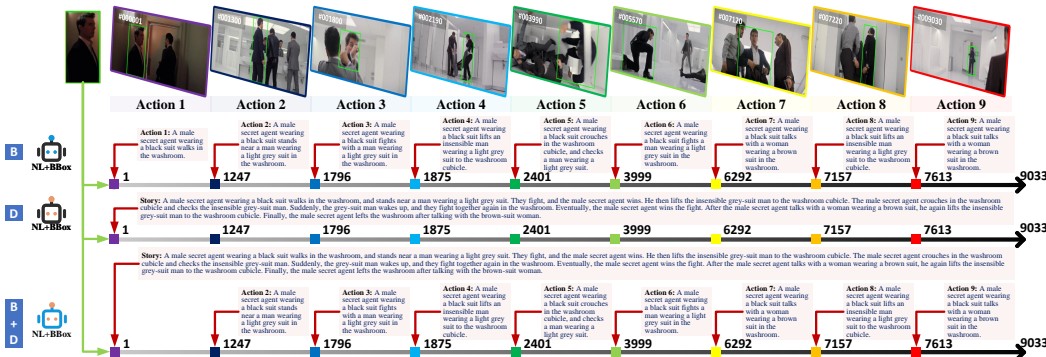

Figure A5: An illustration of mechanism B, mechanism B, and their combination.

Table A5: Results of mechanism B, mechanism B, and their combination.

| Tracker | Architecture | Initialize | Mechanism | PRE | N-PRE | SR |
|---------|-------------|-----------|-----------|-----|-------|-----|
| **SNLT** [46] | SNN | NL&BBox | Action (B) | 0.004 | 0.226 | 0.036 |
| | | | Story (D) | 0.005 | 0.230 | 0.040 |
| | | | Combination (B+D) | 0.004 | 0.229 | 0.037 |
| **VLT_SCAR** [42] | SNN | NL&BBox | Action (B) | 0.116 | 0.354 | 0.167 |
| | | | Story (D) | 0.127 | 0.403 | 0.184 |
| | | | Combination (B+D) | 0.107 | 0.367 | 0.165 |
| **VLT_TT** [42] | Transformer | NL&BBox | Action (B) | 0.318 | 0.602 | 0.468 |
| | | | Story (D) | 0.322 | 0.616 | 0.480 |
| | | | Combination (B+D) | 0.327 | 0.612 | 0.477 |
| **JointNLT** [18] | Transformer | NL&BBox | Action (B) | 0.445 | 0.786 | 0.610 |
| | | | Story (D) | 0.433 | 0.773 | 0.600 |
| | | | Combination (B+D) | 0.443 | 0.783 | 0.607 |

Considering that integrating information from different granularities may further benefit the algorithms, we here take mechanisms B and D as an example to explore whether multiple granularities of information are more effective for algorithms. As shown in Figure A5, we combine mechanisms B and D using the following strategy: taking mechanism B as the main component, we replace its semantic information from the first frame with *story* information (mechanism D). The experimental results are shown in Table A5.

Since most current multi-modal trackers perform target feature matching frame by frame, when the first frame receives *story* information, the semantic information will be replaced by new *action* information after reaching the next *action*. Therefore, the effective range of *story* information only covers the initial *action*. As a result, for most algorithms, the score difference between combination (B+D) and the original mechanism B is insignificant.

It is worth noting that this experiment only provides a simple and direct approach to evaluating tracking performance with the combination of multi-granularity information. The direct reason for insignificant improvement lies in the limitations of existing trackers (lacking a well-designed semantic processing module and the poor multi-modal alignment capability). If future multi-modal tracking algorithms can design a stronger semantic information processing module to comprehensively represent the video content (*i.e.*, hierarchically constructing video content based on graph [43]), perhaps a more powerful tracking capability can be obtained than simply using a single granularity of information.

