# OpenReview forum: "A Multi-modal Global Instance Tracking Benchmark (MGIT): Better Locating Target in Complex Spatio-temporal and Causal Relationship"
_NeurIPS.cc/2023/Track/Datasets_and_Benchmarks — NeurIPS 2023 Datasets and Benchmarks Poster_

### Official Review · Reviewer_u9AZ · 2023-07-20
**Review for A Multi-modal Global Instance Tracking Benchmark (MGIT)**

**Rating:** 8
**Confidence:** 5
**Correctness:** Yes
**Clarity:** Yes

**Strengths:**

1. The paper is well-structured, and the diagrams effectively demonstrate the multi-granular annotations and data statistics of the dataset. Furthermore, the detailed and well-crafted tables underscore the value of the proposed dataset.

2. The paper commendably proposes a challenging multi-modal global instance tracking benchmark with multi-granular annotation.

3. Previous language-based tracking tasks have often been constrained by the precision of the description statement of the object in the first frame, as well as inaccuracies in the language description due to significant state changes of the object in long videos. MGIT addresses these issues by introducing multi-granular annotations, which break down the problem and provide a more accurate language description.

**Additional Feedback:**

Good dataset website.

**Documentation:**

Yes

**Limitations:**

Yes

**Opportunities For Improvement:**

1. Is there a standard to measure the annotation errors from different annotators? It may be valuable to discuss potential issues with ambiguous annotations that might arise with fine-grained annotations.

2. While comprehensive experiments have been conducted on SOTA trackers on the dataset, the comparison with 'bad case' scenarios seems to be lacking. For instance, failures of trackers due to language inaccuracies in other datasets could be discussed. It would be insightful to conduct a visual analysis to understand what specific situations within this dataset could pose significant challenges to the trackers.

**Relation To Prior Work:**

Yes

**Summary And Contributions:**

The authors propose a multi-modal global instance tracking benchmark, MGIT, comprising 150 long video sequences totaling over 2.03 million frames. These sequences aim to encapsulate the intricate spatio-temporal and causal relationships present in extended narrative content.
The authors annotate each video sequence with three semantic grains - action, activity, and story, thus modeling the progressive process of human cognition. They anticipate that this multi-granular annotation strategy will stimulate multi-modal object tracking research and long video understanding.
Comparative experiments on existing multi-modal object tracking benchmarks were conducted to explore the effects of various annotation methods and validate the viability of the proposed annotation method.

---

> ### Author Response · Authors · 2023-08-21
> **Thanks to Reviewer u9AZ and Our Reply**
>
> Thanks to the positive comments and meaningful suggestions. We summarize questions and list answers here:
>
> **Q1. Is there a standard to measure the annotation errors from different annotators?**
>
> A1. It is challenging to establish a specific metric for measuring the quality of semantic annotations, and previous representative works (such as OTB_Lang, LaSOT, TNL2k) have not specifically designed or tested their semantic annotations. Therefore, we do not design a metric to quantify the semantic annotations, but we have implemented numerous strategies to ensure annotation quality. (1) **Task Decomposition**: We first decompose the task to ensure a standardized workflow for execution. For instance, we begin by annotating the finest granularity (action), and subsequently continue with the annotations of activity and story. This approach ensures accuracy and consistency in the fundamental content throughout various levels of granularity. Given that action is the finest granularity and its annotation quality may affect activity and story, we refer to film narrative literature and English grammar materials to further decompose the description of an action. This decomposition involves identifying the tracking target (who), the motion (what), the presence of a third-party object (if applicable), the location (where), and the time interval (when). By obtaining these specific details, annotators can attain a standardized and comprehensive description of the action. (2) **Annotator Selection**: Considering the difficulty of controlling annotation quality in crowdsourcing, we chose highly cognitive graduate students with experience in dataset annotation to form an annotation team. Team members not only have experience in annotating vision-based datasets represented by VideoCube but also have experience in annotating image datasets in visual psychology. They have a solid foundation in dataset construction in fields such as computer vision and cognitive psychology. All team members undergo standardized training before formal annotation to ensure their understanding of task characteristics and annotation rules. Additionally, the training session includes 10 video examples of different types, requiring annotators to comprehend the annotation process and details. (3) **Annotation Workflow**: The formal annotation process begins, wherein annotation personnel is grouped based on video types, including cartoons, movies, TV shows, sports, performances, and documentaries. Any issues requiring discussion will be documented, followed by a comprehensive discussion among all personnel, and then the annotation process for that particular sequence will commence. Furthermore, to enhance the standardization of annotations, we refer to WordNet to construct verb and noun lists.  Initially, annotators will choose candidate terms from the current vocabulary lists to depict the essential elements of the scene, aiming to maintain consistency in the portrayal of actions across varying sequences to the greatest extent possible. If there are no appropriate terms found in the candidate list, annotators will employ new vocabulary to depict the elements and subsequently incorporate them into the candidate list, supplemented with relevant examples for future annotation reference.
> (4) **Quality Review**: After completing the annotation for all sequences, we will review the content to ensure its quality. Additionally, we utilize the Stanford CoreNLP, a natural language processing toolkit, to examine the semantic annotations and ensure the grammatical structure's completeness. This information has been included in the Appendix (Section A.3.2). Thanks.
>
> **Q2. While comprehensive experiments have been conducted on SOTA trackers on the dataset, the comparison with 'bad case' scenarios seems to be lacking.**
>
> A2. Following your suggestion, we have added Section 4.4 of the revised version and introduced Figure 7 to showcase four bad cases. The first two examples are selected from LaSOT, demonstrating that ambiguous semantic information may introduce noise, leading algorithms to focus on similar objects wrongly. This emphasizes the importance of accurate semantic annotations. The latter two examples are chosen from MGIT, demonstrating that the experimental environment constructed by MGIT presents complex spatio-temporal and causal relationships, posing challenges to multi-modal algorithms. Additionally, using only the first action information (mechanism A) is applied in all other multi-modal SOT benchmarks. However, it is not applicable to visual object tracking in complex scenes like MGIT. Therefore, the multi-granularity semantic annotations provided in our work offer a more reasonable solution. Based on this, we conduct a detailed analysis of the reasons for algorithm failures and identify their bottlenecks. Thanks.

---

### Official Review · Reviewer_ZBXv · 2023-07-21
**Good paper with sufficient details**

**Rating:** 7
**Confidence:** 4
**Clarity:** This paper is well written. The idea …

**Strengths:**

1.	This paper contributes a new benchmark for multi-modal tracking (MGIT). It consists of 150 long video sequences with a total of 2.03 million frames. Each video sequence is annotated with three semantic grains (i.e., action, activity, and story).

2.	The authors execute comparative experiments to explore the impact of different annotation methods and validate that the proposed annotation method is a feasible solution.

3.	Detailed experimental analyses are conducted on MGIT, supporting further research in multi-modal object tracking.

4.	The motivation for dataset construction is strong and well-presented. The logic and structure of the paper are clear, and the quality of this benchmark is further enhanced by sufficient analysis. This work can help researchers to conduct further research in object tracking and video understanding.


**Additional Feedback:**

See Opportunities For Improvement

**Correctness:**

The dataset is constructed in a sound way. The evaluation methods and experiments are performed correctly.

**Documentation:**

Sufficient details are provided, including the data collection and organization, availability and maintenance, and ethical and responsible use. Sufficient details support reproducibility for the readers.

**Ethics:**

See Opportunities For Improvement

**Limitations:**

The authors address the limitations of existing works by introducing a new benchmark for multi-modal tracking. They are upfront about the limitations of their work. However, the experiment in Table 2 is very confusing. The results of SNLT are too worse. Natural language knowledge should be auxiliary information that can improve the performance of the tracker. However, the results of the SNLT show that natural language knowledge completely impairs the effect of tracking, and the accuracy is extremely low. The results require further explanation.

**Opportunities For Improvement:**

1.	It is better to describe the evaluation methods with specific formulas.

2.	The performances of mechanisms B, C, and D are reduced. The different key challenges of tracking with three annotations (action, activity, and story) should be further presented.

3.	Whether these three annotations can be combined (D+B) to improve the tracking performance？ The authors need to conduct further consideration and exploration.


**Relation To Prior Work:**

The authors have properly discussed the differences and relations to previous works. It is a good paper.

**Summary And Contributions:**

1.	This paper proposes a new multi-modal benchmark named MGIT.
2.	A multi-granular annotation strategy is designed for providing scientific semantic information.
3.	This paper executes comparative experiments on other benchmarks.
4.	The authors conduct detailed experimental analyses on MGIT.

---

> ### Author Response · Authors · 2023-08-21
> **Thanks to Reviewer ZBXv and Our Reply**
>
> Thanks to the positive comments and meaningful suggestions. We summarize questions and list answers here:
>
> **Q1. Add evaluation methods with specific formulas.**
>
> A1. We introduce the calculation process in the main text (Section 4.1) and add the formulas to the Appendix (Section B.1). Thanks.
>
> **Q2. The key challenges of tracking should be presented.**
>
> A2. We discuss from two perspectives: (1) **Comparison of different multi-modal evaluation mechanisms**: First, compared to mechanism A, which only provides semantic information for the first action, most multi-modal methods do not show a significant performance decrease in mechanisms B, C, and D. Except JointNLT, the success rate scores of other multi-modal trackers in mechanism D are superior to mechanism A. Furthermore, most algorithms performed the worst in mechanism A because they only obtained semantic descriptions for the first action, and this semantic information is not updated in the subsequent process, which may introduce noise to the tracker as the sequence progresses. Mechanisms B and C both regularly update the semantic information during tracking. However, C has a moderate frequency of semantic information updates, and each update provides a moderate length of semantic information, which may better leverage the capabilities of trackers. Mechanism D, like A, only provides semantic information in the initial frame but offers story information that can cover the entire video. However, current multi-modal trackers lack well-designed semantic understanding modules for handling long texts, making it challenging to align semantic information with visual information. (2) **Comparison between single-modal and multi-modal approaches**: Semantic modalities can provide information beyond superficial features such as appearance and location compared to pure visual trackers. However, achieving better correlation and fusion between modalities still needs to be solved. Previous research on other multi-modal tasks has demonstrated that multi-modal approaches can introduce more information, resulting in improved algorithms (What Makes Multi-modal Learning Better than Single (Provably), NeurIPS21). However, for SOT research, the performance of multi-modal algorithms still lags behind that of single-modal algorithms. The main reason may be the need for more high-quality benchmarks -- existing multi-modal benchmarks have significant limitations in the completeness of semantic information and video complexity, making it challenging to provide a favorable experimental environment. Additionally, these benchmarks do not adopt a multi-granular annotation strategy, resulting in an evaluation system that only involves mechanism A. MGIT can provide a high-quality experimental environment for research, and help the algorithms quickly identify bottleneck issues under various mechanisms, thereby accelerating the development of multi-modal trackers. This information has been included in the Appendix (Section B.3). Thanks.
>
> **Q3. Performance of combining B+D.**
>
> A3. We take mechanism B as the main component and replace its semantic information from the first frame with story-level (D).
>
> |Tracker|Mechanism|SR|
> |----|-----|---|
> |SNLT| B|0.036|
> ||D|0.040|
> ||B+D|0.037|
> |VLT_SCAR|B|0.167|
> ||D|0.184|
> ||B+D|0.165|
> |VLT_TT|B|0.468|
> ||D|0.480 |
> ||B+D|0.477|
> |JointNLT|B|0.610|
> ||D|0.600|
> ||B+D|0.607|
>
> When the first frame receives story-level information, the semantic information will be replaced by new action information after reaching the next action. Therefore, the effective range of story-level information only covers the initial action. Thus, for most algorithms, the score difference between B+D and the original B is insignificant. More discussion has been included in the Appendix (Section B.4). Thanks.
>
> **Q4. Why SNLT performs so worse?**
>
> A4. Possible reasons are as follows: (1) SNLT (CVPR21) is based on the local search, which exhibits a performance gap to the global search trackers. Experiments reveal that local search trackers may encounter a more severe tracking drift problem in the MGIT task (this method tracks by cutting out the search area from the original image, while the high image resolution in MGIT will challenge it). Besides, SNLT's weaker tracking ability is more prone to failure. The errors generated as a result will further misguide, thus creating a negative loop. For example, when it loses the target or drifts towards a similar object, it will persist in tracking failure until the target reemerges within its search area with more recognizable visual information. (2) There are some issues within SNLT's open-source code, and although we have fixed them, SNLT has some limitations compared to other trackers, which may also cause to its poor performance. This information is included in the Appendix (Section B.3). Thanks.

---

### Official Review · Reviewer_uBXK · 2023-07-21
**A Useful multi-modal global instance tracking benchmark.**

**Rating:** 7
**Confidence:** 4
**Correctness:** I do not have any concerns about the …
**Clarity:** The paper is well written.

**Strengths:**

1. This paper proposed a challenging multi-modal global instance tracking benchmark.
2. The multi-granular annotation strategy is interesting and can provide more dense semantic descriptions for multi-modal GIT task. Such a strategy can provide a step-by-step learning power like human being for multi-modal trackers.
3. Previous language-based tracking tasks often suffered from problems such as semantic ambiguity or incomplete description of the overall state of the sequence objects. However, the introduction of MGIT has brought about new paradigms and trends in addressing long-term multi-modal tracking, which is highly encouraging.
4. This article is written fluently and smoothly. I can read it very well. The exquisite illustrations help me better understand the processes of data collection, natural language annotation, experiments, etc.
5. In different multimodal benchmark test results, it is shown that MGIT is more challenging, and a scientific annotation strategy can avoid the interference of ambiguous information. Meanwhile, comparisons are also made with newer types of transform trackers.


**Additional Feedback:**

No

**Documentation:**

Dataset is well documented.

**Ethics:**

I do see any ethical concerns about this paper.

**Limitations:**

1. The author did not specify who is responsible for annotating natural language and dividing motion, third-party, and location in actions. If it is done by a crowdsourcing team, has the author employed any means to ensure the accuracy of semantic descriptions and avoid semantic ambiguity caused by improper division?
2. If the author could design other meters to evaluate these three mechanisms, it would make the construction of MGIT more convincing.


**Opportunities For Improvement:**

Currently, MGIT includes six types of topics. In the future, if more types can be added, it will contribute to the development of data-driven multi-modal tracking models.

**Relation To Prior Work:**

The contribution of this work compared to previous art is clearly discussed.

**Summary And Contributions:**

This paper proposed a new multi-modal global instance tracking benchmark named MGIT, which consists of 150 long vides with a total of 2.03M frames. This paper designed a new multi-granular annotation strategy for providing scientific semantic information, from action, activity to story. The detailed experimental analyses on MGIT indicate that existing methods still have significant room for improvement in multi-modal tracking.

---

> ### Author Response · Authors · 2023-08-21
> **Thanks to Reviewer uBXK and Our Reply**
>
> Thanks to the positive comments and meaningful suggestions. We summarize questions and list answers here:
>
> **Q1. Currently, MGIT includes six types of topics. In the future, if more types can be added, it will contribute to the development of data-driven multi-modal tracking models.**
>
> A1. We selected six video categories in this work since they have covered most of the video types encountered daily (e.g., video websites), making them representative. In future work, we will carefully consider your suggestions and include more video categories to further serve the advancement of the research field. Thanks.
>
> **Q2. The author did not specify who is responsible for annotating natural language and dividing motion, third-party, and location in actions. If it is done by a crowdsourcing team, has the author employed any means to ensure the accuracy of semantic descriptions and avoid semantic ambiguity caused by improper division?**
>
> A2. We chose an elite annotation team instead of crowdsourcing to carry out this work and ensured quality through multiple efforts. (1) **Task Decomposition**: We first decompose the task to ensure a standardized workflow for execution. For instance, we begin by annotating the finest granularity (action), and subsequently continue with the annotations of activity and story.  This approach ensures accuracy and consistency in the fundamental content throughout various levels of granularity. Given that action is the finest granularity and its annotation quality may affect activity and story, we refer to film narrative literature and English grammar materials to further decompose the description of an action. This decomposition involves identifying the tracking target (who), the motion (what), the presence of a third-party object (if applicable), the location (where), and the time interval (when). By obtaining these specific details, annotators can attain a standardized and comprehensive description of the action. (2) **Annotator Selection**: Considering the difficulty of controlling annotation quality in crowdsourcing, we chose highly cognitive graduate students with experience in dataset annotation to form an annotation team. Team members not only have experience in annotating vision-based datasets represented by VideoCube but also have experience in annotating image datasets in visual psychology. They have a solid foundation in dataset construction in fields such as computer vision and cognitive psychology. All team members undergo standardized training before formal annotation to ensure their understanding of task characteristics and annotation rules. Additionally, the training session includes 10 video examples of different types, requiring annotators to comprehend the annotation process and details. (3) **Annotation Workflow**: The formal annotation process begins, wherein annotation personnel is grouped based on video types, including cartoons, movies, TV shows, sports, performances, and documentaries. Any issues requiring discussion will be documented, followed by a comprehensive discussion among all personnel, and then the annotation process for that particular sequence will commence. Furthermore, to enhance the standardization of annotations, we refer to WordNet to construct verb and noun lists.  Initially, annotators will choose candidate terms from the current vocabulary lists to depict the essential elements of the scene, aiming to maintain consistency in the portrayal of actions across varying sequences to the greatest extent possible. If there are no appropriate terms found in the candidate list, annotators will employ new vocabulary to depict the elements and subsequently incorporate them into the candidate list, supplemented with relevant examples for future annotation reference. (4) **Quality Review**: After completing the annotation for all sequences, we will review the content to ensure its quality. Additionally, we utilize the Stanford CoreNLP, a natural language processing toolkit, to examine the semantic annotations and ensure the grammatical structure's completeness. This information has been included in the Appendix (Section A.3.2). Thanks.
>
> **Q3. If the author could design other meters to evaluate these three mechanisms, it would make the construction of MGIT more convincing.**
>
> A3. A well-design evaluation system is crucial for drawing experimental conclusions. We mainly focus on the characteristics of the task and innovate existing evaluation systems at the level of experimental mechanisms (mechanisms B, C, D). However, we adopt the classic evaluation metrics for the SOT task, aiming to help readers easily understand the meaning behind the result scores and concentrate on analyzing the performance under different granularities of semantic information. In future work, we will carefully consider your suggestions and innovate evaluation metrics to achieve a more comprehensive analysis of algorithms. Thanks.

---

### Official Review · Reviewer_dttb · 2023-07-21
**Interesting work but with technical details unclear**

**Rating:** 6
**Confidence:** 5
**Correctness:** The submitted dataset is constructed …
**Clarity:** This paper is well-written.

**Strengths:**

(1) This paper provides a complex dataset with long-sequences video segments, which is clearly a more challenging task.
(2) This paper proposes a multi-granular annotation strategy, which can be divided into three parts, i.e., action, activity, and story. This strategy aims at coupling human understanding into semantic labels.
(3) This dataset is challenging for previous models, which perform poorly on this dataset.

**Additional Feedback:**

Please refer "Opportunities For Improvement" part.

**Documentation:**

Somewhat sufficient

**Limitations:**

Overall, there are some technical details unclear in current version as described above.

**Opportunities For Improvement:**

(1) All the 150 video segments are chose from a former work named Videocube [1], which has 500 videos in total. MGIT has 150 videos, and 2.03 million frames (L.11-13). The average frame for each video in MGIT is around 13K, which is quite similar to the average frame for each video in Videocube (14K). Obviously you do not select videos from Videocube simply by frame number, so what is your strategy in video selecting? In other words, what is your contribution in video selecting part?
(2) This paper points out that most of the existing datasets has the problem of semantic ambiguity. Though some cases are shown, there is not enough experiments or statistics to support this idea.
(3) This paper claims that the proposed multi-granular annotation strategy helps couple human understanding into semantic labels, and shows the corresponding experiment in Sec.4.2.
 a)However, in this section, the paper tries to prove the effectiveness of their proposed annotation strategy by comparing the performances on other datasets, i.e., LaSOT_{sub} and LaSOT_{NLC}. In other words, it is obviously clear that models can achieve better performance on a dataset without semantic ambiguity, but how can you show the necessity of your annotation strategy?
 b) Furthermore, some baseline methods perform worse using the story annotation than using the action or activity annotation. With such fact, what is the necessity of introducing the annotation strategy?
 c) The main contribution lies in this part, but as I mentioned above, I still cannot see the effectiveness and necessity of the proposed annotation strategy.
(4) There are 101 zip and other format files in your Train Data link provided in the Appendix. Quite hard to understand without a instruction.
(5) The main contribution of this paper (the annotation strategy) is relatively limited and is not proven valid.
(6) The motivation is quite close to NExT-QA [2], which is not discussed in this paper.
[1] Global Instance Tracking: Locating Target More Like Humans. T-PAMI 2022.
[2] Video as Conditional Graph Hierarchy for Multi-Granular Question Answering. AAAI 2022.

**Relation To Prior Work:**

The motivation is quite close to NExT-QA, which is not discussed in this paper.

**Summary And Contributions:**

This paper proposes a multi-modal global instance tracking dataset named MGIT, which includes complicated video segments and multi-granular annotations. Experimental results show that existing methods perform poorly on this dataset, and still have significant room for improvement.

---

> ### Author Response · Authors · 2023-08-21
> **Thanks to Reviewer dttb and Our Reply**
>
> Thanks to the meaningful suggestions. We summarize questions and list answers here:
>
> **Q1. Strategy in video selecting.**
>
> A1. We assess the similarity of MGIT and VideoCube across various dimensions, including difficulty-level. Three SOTA trackers (MixFormer, KeepTrack, SiamRCNN) are selected, and their success scores are ranked to measure difficulty. By considering both the distribution across 6D principle and difficulty-level, we choose 150 videos to construct MGIT. This content has been added in the Appendix (Section A.2). Thanks.
>
> **Q2. Lack statistics for existing datasets.**
>
> A2. We conduct statistical analyses on representative datasets from two aspects: (1) Ambiguity: 10 sequences are randomly selected from each dataset for inspection. The pass rates are 30% (OTB-Lang), 70% (LaSOT), 60% (LaSOT-ext), and 60% (TNL2k). (2) Completeness: Semantic descriptions should have complete grammatical structures. Hence, Stanford CoreNLP is utilized to count complete sentences. The pass rates are 9% (OTB-Lang), 63% (LaSOT), 36% (LaSOT-ext), and 20% (TNL2k). Thus, existing datasets exhibit deficiencies in ambiguity and completeness. This part has been added in Appendix (Section A.3.1). Thanks.
>
> **Q3. The necessity of the annotation strategy.**
>
> A3. We make efforts to ensure no ambiguity and validate its necessity through experiments. (1) High-quality annotation: We obey a strict process to exclude ambiguity. It includes four steps (task decomposition, annotator selection, annotation workflow, and quality review) and has been updated in Section A.3.2 of the Appendix. (2) Experimental analysis: With accurate semantic annotation, richer semantic information (mechanism D) should improve the performance more than a simple sentence (mechanism A). Results show that most trackers perform well under D than A. The only JointNLT that does not show this phenomenon is due to its inability to handle story-level input (exceeding 50 tokens), but it obtains higher scores on both B (richer action information) and C (richer activity information), which also shows that high-quality semantic label does help tracker to improve performance. It also confirms that this work can identify the bottlenecks -- trackers should improve their ability to process long texts and align multi-modal information. Thus, the multi-granular semantic information provided by MGIT will help them better locate target in long videos. Above analysis has been added into Section 4.3 of the revision to clarify the contributions. Thanks.
>
> **Q4. The format of zip files.**
>
> A4. Long videos are divided into smaller segments (under 4 GB) to facilitate downloading. The decompression scripts and instructions are provided on the website. This information has been included in the Appendix (Section A.4). Thanks.
>
> **Q5. Concerns about the main contribution.**
>
> A5. Analyses in A2 and A3 have explained contributions from the limitations of existing benchmarks and the necessity of the proposed annotation strategy. We have updated the revision to further clear the main contributions. Thanks.
>
> **Q6. Discussion with NExT-QA.**
>
> A6. We use HQGA to represent the method (AAAI22) and NExt-QA to indicate the dataset (CVPR21). We compare MGIT with them and summarize the similarities and differences: (1) **Similarities**: HQGA proposes a VQA method for multi-granularity analysis. Its hierarchical modeling idea and our work have similarities in the modeling object – human comprehension ability. Thus, the sameness verifies the scientific of the proposed multi-granularity annotation strategy. (2)**Differences**: (a) Tasks: NExT-QA and HQGA concentrate on the VQA task. The question-answer pairs in VQA task are usually centered around the interactivity between multiple subjects in a restricted space, thus focusing more on spatial complexity. In contrast, the GIT task focuses on the motion of a single target in complex spatio-temporal relationships. (b) Video contents: Because VQA focuses more on spatial relations, the average duration of NExT-QA is relatively short (1.3k frames in average length at 30FPS). In contrast, MGIT pays more attention to the spatio-temporal and causal relationships in longer sequence, and its average length is 10 times longer than NExT-QA. (3) Granularity divisions: Due to the limited complexity of the temporal relationship, the highest level of HQGA matches the action-level or the activity-level of MGIT, but cannot be aligned with the challenging contained in story-level.
> In summary, HQGA provides a hierarchical analysis framework for the VQA task, and mainly focuses on modeling the interaction relationship of multiple subjects in a scene. In contrast, our work focuses on tracking the moving object under spatial-temporal variations across different scenes. Thus, their motivations are different. Besides, we have cited these two articles in the revision paper (Section 3.2) as examples of decoupling video content based on the human-like hierarchical structure. Thanks.

---

### Author Response · Authors · 2023-08-21
**Rebuttal Summary**

We especially appreciate all the reviewers for their invaluable comments and suggestions. Most reviewers believe this work is interesting and well-written, acknowledging its potential to advance future research. Furthermore, we have thoroughly addressed all the concerns the reviewers raised and made the following updates in our revision. The adaptations in our revision are as follows:

(1)	**Dataset construction (Section 3)**: We add detailed information about the data selection principle and semantic annotation process to show our efforts in guaranteeing quality. Additionally, we have included an introduction to relevant methods in other fields and further analyzed the limitations of existing multi-modal SOT benchmarks in semantic annotation -- these serve to further illustrate our motivation.

(2)	**Evaluation Methods (Section 4.1)**: We provide relevant computational formulas to assist readers in understanding the calculation process and the meaning of final results.

(3)	**Explanation about results (Section 4.3)**: We conduct further analysis on the experimental results and delve into the limitations of existing methods.

(4)	**Visualization and Bad Case Analysis (Section 4.4)**: We add four bad cases to illustrate the limitations of existing benchmarks and the challenges of MGIT.

In addition, due to space limitations in the main text, more updated details and more thorough discussions have been documented in the Appendix. Please refer to the revised version for detailed information; all the updates have been highlighted in blue font.

---

### Decision · Program_Chairs · 2023-09-22

**Decision:**

Accept (Poster)

**Comment:**

The authors present MGIT, a multi-modal benchmark for long video understanding. It comprises 150 sequences with over 2 million frames annotated with multi-granular semantics - action, activity, and story. This mimics human cognition and aims to address issues with previous benchmarks and advance multi-modal tracking and video understanding. Detailed experiments on current methods demonstrate the challenging nature of this benchmark and validate the usefulness of this dataset to the field.

Pros:
- High-quality, challenging multimodal video benchmark to advance tracking and video understanding consisting of much longer videos than previous benchmarks.
- Proposes multi-granular annotation approach (action, activity and story) addressing issues with previous language-based tracking benchmarks.
- Provides extensive evaluation of existing SOTA tracking models demonstrating large room for improvements and stimulating new research.
- Well written and structured paper with helpful illustrations and easy to parse tables. Detailed and helpful appendix incorporating feedback by the reviewers.

Cons:
- (Minor) As mentioned in the paper’s Limitations section adding additional video categories could improve the usefulness of the dataset even further.

Unanimous support by the reviewers in favor of accept with two good or higher and two borderline (one rating was increased after authors addressed concerns). AC agrees with the reviewers and recommends Accept.